# CFTR Rescue by Lumacaftor (VX-809) Induces an Extensive Reorganization of Mitochondria in the Cystic Fibrosis Bronchial Epithelium

**DOI:** 10.3390/cells11121938

**Published:** 2022-06-16

**Authors:** Clarissa Braccia, Josie A. Christopher, Oliver M. Crook, Lisa M. Breckels, Rayner M. L. Queiroz, Nara Liessi, Valeria Tomati, Valeria Capurro, Tiziano Bandiera, Simona Baldassari, Nicoletta Pedemonte, Kathryn S. Lilley, Andrea Armirotti

**Affiliations:** 1D3 PharmaChemistry, Istituto Italiano di Tecnologia, Via Morego 30, 16163 Genova, Italy; clarissa.braccia@iit.it (C.B.); tiziano.bandiera@iit.it (T.B.); 2Cambridge Centre for Proteomics, Department of Biochemistry, University of Cambridge, Tennis Court Road, Cambridge CB2 1QR, UK; jac290@cam.ac.uk (J.A.C.); oliver.crook@stats.ox.ac.uk (O.M.C.); lms79@cam.ac.uk (L.M.B.); raynermyr@gmail.com (R.M.L.Q.); 3Department of Statistics, University of Oxford, 29 St Giles’, Oxford OX1 3LB, UK; 4Analytical Chemistry Facility, Istituto Italiano di Tecnologia, Via Morego 30, 16163 Genova, Italy; nara.liessi@iit.it; 5UOC Genetica Medica, IRCCS Istituto Giannina Gaslini, Via Gerolamo Gaslini 5, 16147 Genova, Italy; valeria.tomati@libero.it (V.T.); valeriacapurro@gaslini.org (V.C.); simonabaldassari@gaslini.org (S.B.)

**Keywords:** Lumacaftor, mitochondria, peroxisomes, spatial proteomics, primary cells

## Abstract

Background: Cystic Fibrosis (CF) is a genetic disorder affecting around 1 in every 3000 newborns. In the most common mutation, F508del, the defective anion channel, CFTR, is prevented from reaching the plasma membrane (PM) by the quality check control of the cell. Little is known about how CFTR pharmacological rescue impacts the cell proteome. Methods: We used high-resolution mass spectrometry, differential ultracentrifugation, machine learning and bioinformatics to investigate both changes in the expression and localization of the human bronchial epithelium CF model (F508del-CFTR CFBE41o-) proteome following treatment with VX-809 (Lumacaftor), a drug able to improve the trafficking of CFTR. Results: The data suggested no stark changes in protein expression, yet subtle localization changes of proteins of the mitochondria and peroxisomes were detected. We then used high-content confocal microscopy to further investigate the morphological and compositional changes of peroxisomes and mitochondria under these conditions, as well as in patient-derived primary cells. We profiled several thousand proteins and we determined the subcellular localization data for around 5000 of them using the LOPIT-DC spatial proteomics protocol. Conclusions: We observed that treatment with VX-809 induces extensive structural and functional remodelling of mitochondria and peroxisomes that resemble the phenotype of healthy cells. Our data suggest additional rescue mechanisms of VX-809 beyond the correction of aberrant folding of F508del-CFTR and subsequent trafficking to the PM.

## 1. Introduction

Cystic Fibrosis (CF) is a genetic autosomal recessive disorder that affects approximately 1 every 3000 births (Cystic Fibrosis Foundation, “Patient Registry Annual Data Report”, 2018). CF is caused by mutations of the gene encoding for the cystic fibrosis transmembrane conductance regulator protein (CFTR), a transmembrane anion channel expressed at the apical membrane of epithelial cells of the airway, the gastrointestinal tract, the pancreas, and the biliary and sweat ducts. CFTR has an important role in the regulation of chloride and bicarbonate ion transport and water homeostasis. Its mutations result in dysfunctional ion transport across the apical membrane at the surface of several epithelia, generating thickened and dehydrated secretions. In the lung, this leads to a decrease in the mucociliary clearance, favoring bacterial colonization and progressive obstruction of the duct. Although over 2000 CFTR variants have been identified so far, the most common mutation is a deletion of the phenylalanine in position 508 (F508del), which shows an allelic frequency of around 90% among CF patients [1]. F508del-CFTR is incorrectly folded, causing its retention at the endoplasmic reticulum (ER) and subsequent proteasomal degradation. Moreover, the small amount of mutant CFTR that reaches the membrane is unstable and shows gating defects [2].

### 1.1. CFTR Biogenesis

In healthy cells, CFTR biogenesis begins in the endoplasmic reticulum (ER): the protein undergoes co-translational folding into the ER membrane [3] and subsequent N-glycosylation at the Asn894 and Asn900 residues in the ER lumen [4]. This glycosylation is critical for the ER quality control (ERQC) process that monitors the correct folding status of proteins before they exit the ER [5]. The glycosylated immature form of wild type (wt)-CFTR traffics to the Golgi complex, where it is processed by several Golgi glycosyltransferases to create the fully mature form of wt-CFTR [6]. Mature CFTR exits the trans-Golgi network (TGN) in vesicles as part of the intracellular protein transport system to the plasma membrane (PM) [7]. CFTR insertion into the PM is driven by myosin myo5b, Ras-related protein Rab 11 and the PDZ adaptor NHERF1 [8]. CFTR is then tethered to the PM through the actin cytoskeleton. Several proteins [9], at different checkpoints, recognize the defective folding of F508del-CFTR and target it for degradation [10]. At the PM, misfolded CFTR is rapidly ubiquitinated and trafficked to endosomal compartments [11], to be either deubiquitinated, refolded at the ER and recycled to the PM or trafficked to the lysosome for degradation [11]. Marked levels of CFTR misfolding and increased CFTR degradation are a phenotype of several CFTR mutations, including the most common mutation, F508del.

### 1.2. Protein Alterations in CF

At the basic mechanistic level, CF is thus a trafficking disease for many of the known mutations. The role of protein expression and post-translational modifications in CF has been extensively investigated in the past [12,13], but, to date, no studies report if the pharmacological rescue of CFTR is associated with changes in the global proteome of the cell. Moreover, to our knowledge, no papers focused on mapping the effects of VX-809 (Lumacaftor^®^) on the subcellular proteome. VX-809 is a drug that partially corrects the aberrant folding of F508-del CFTR by stabilizing one of its membrane-spanning domains [14], subsequently suppressing the misfolding defects by improving intramolecular interactions within CFTR, allowing evasion of quality control checkpoints and aiding its localization to the PM [15]. Here, we investigated if the exposure to VX-809 alters the proteome of the human bronchial epithelium by using the CFBE41o- cell model, stably over-expressing F508del-CFTR (F508del-CFTR CFBE41o-). First, we profiled the global F508del-CFTR CFBE41o- proteome, looking for changes in protein expression. Secondly, we analyzed the subcellular localization of proteins on both a global and targeted scale and we explored how it is impacted by F508del-CFTR rescue triggered by VX-809. To achieve a subcellular map of the proteome, we employed the Localization of Organelle Proteins by Isotope Tagging after Differential ultra-Centrifugation (LOPIT-DC) [16] approach, a powerful and well-established tool for the investigation of the spatial distribution of global proteome within cells, which combines subcellular fractionation based on differential centrifugation, quantitative mass spectrometry, and machine learning. Its application provides a global snapshot of the steady-state subcellular distribution of proteins. In this study, we aimed at identifying proteins that change their expression and/or cellular localization with VX-809, to uncover molecules that possibly play a role in the trafficking of the defective CFTR to the PM. These molecules might not be direct interactors of CFTR but they might be involved in biological processes that promote the trafficking of the misfolded protein at the PM [17,18,19] and/or that reduce its cellular recycling through ubiquitin-controlled quality check mechanisms [9,20,21]. The identification of these proteins will assist in further dissecting the CF pathophysiological mechanisms and help in the quest for new pharmacological targets. We then verified and further investigated whether the effects observed from the LOPIT-DC data are specifically linked to CFTR rescue by performing high-content quantitative imaging on parental and wildtype-CFTR (wt-CFTR) expressing CFBE41o-. Finally, to strengthen these findings, we conducted further imaging on primary bronchial epithelial cells, obtained from CF and non-CF individuals. Indeed, our work highlighted significant alterations in the subcellular proteome in association with VX-809, with a particularly enhanced re-modelling of mitochondrial proteins, that was associated with relevant morphological and functional changes.

## 2. Results

### 2.1. Relative Protein Expression Data

We first analyzed the total cell lysate of the F508delCFTR-CFBE41o- cell model, exposed to either DMSO 0.1% or 1 μM VX-809 (*n* = 5, 24 h), by using the SWATH proteomics protocol we previously published [22]. An additional quality control group (QC), consisting of pools of all the ten samples was also prepared. With this experiment, we quantified the relative levels of expression of 3432 proteins (Appendix A) in each of the samples. We then used Principal Component Analysis (PCA) to explore the obtained dataspace, with the aim to highlight possible significant changes between the experimental groups (DMSO, VX-809 and QC). As reported in Appendix A, while, as expected, all the QC samples are tightly clustered in the middle of the plot, no significant group separation between cells treated with VX-809 or control 0.1% DMSO was observed (Panel A). We then compared the changes in protein expression, employing a univariate statistical test (Student’s *t*-test). After correction for multiple testing (Bonferroni correction), no significantly altered proteins were found (Panel B). We then concluded that 24-h treatment with VX-809 does not induce dramatic changes in the overall protein abundance profiles of the F508del-CFTR CFBE41o- cell model. Moreover, there was insufficient evidence that CFTR abundance is significantly altered by 24-h treatment with VX-809 (Appendix A, Panel C).

### 2.2. LOPIT-DC for F508del-CFTR CFBE41o- Cells

As F508del-mutant CF is considered a trafficking disease, we turned our attention toward the subcellular localization of the F508del-CFTR CFBE41o- proteome, by performing LOPIT-DC. This was to assess whether there may be subcellular changes that otherwise may be overlooked in assays measuring expression. We optimized the published method for the CFBE41o- cell line by adjusting the clearance and number of passages through the ball-bearing homogenizer to achieve optimal lysis and maintenance of organelle membrane integrity [16]. LOPIT-DC relies on the evidence that proteins from the same subcellular compartment show the same abundance profile across the fractions collected by differential centrifugation [23]. Figure 1 summarizes the main steps of LOPIT-DC workflow and how it was applied to investigate the rescue of F508del-CFTR by VX-809 in the CFBE41o- model.

### 2.3. The CFBE41o- Subcellular Proteome before and after VX-809 Treatment

Subcellular proteomics data of F508del-CFTR CFBE41o- cells were collected in biological triplicate, with each individual replicate of control (DMSO-treated) and VX-809-treated samples prepared simultaneously to account for confounding factors. In three individual biological replicates of CFBE41o- cultures treated with DMSO as control, we detected 4842 and 4882 proteins consistently across the triplicates of the control and VX-809-treated samples, respectively. Appendix A summarizes the number of proteins identified in each replicate for both conditions, and Appendix A represents the Venn diagram of the number of proteins detected in both and each of the two conditions for each of the three replicates. The majority of the accessible proteome (4368 proteins) was observed in both conditions. We used a set of 548 manually curated marker proteins (Appendix A), known to localize in specific organelles, as references to pinpoint eleven subcellular localizations: cytosol (58 markers), ER (96), proteasome (29), Golgi (16), lysosomes (17), mitochondria (101), nucleus (107), nucleus–chromatin (32), ribosome (47), peroxisomes (15) and PM(30). The list of organelle markers was created using information available from literature, Uniprot and Gene Ontology databases [23]. The data was visualized using Principal Component Analysis (PCA), generating the first cell-wide subcellular protein map of the CFBE41o- cells, following treatment with control or VX-809, as reported in Figure 2A. A support vector machine (SVM) classifier [24] was used to predict the localization of unassigned proteins to one of the 11 subcellular classes (Figure 2B), applying 5% FDR filtering as described in the pRoloc workflow [25]. As reported in Appendix A, the nucleus and proteasome markers are visibly resolved by comparing different combinations of Principle Components (PC) (PC1 vs. PC3 and PC2 vs. PC4). The separation of Golgi and PM is apparent when comparing PC2 and PC4. Appendix A report the list of all the identified proteins in the three control replicates for the incubation with DMSO and VX-809, respectively, along with their distribution of relative abundance across all organelle fractions and SVM classification data for each protein. The full LOPIT-DC dataset has been made available using a dedicated R Shiny web application: https://proteome.shinyapps.io/cflopit2021/ (accessed on 30 July 2021).

### 2.4. Identification of Differential Localized Proteins

We then focused on finding proteins which showed differential subcellular localization following treatment with VX-809 using a Bayesian non-parametric two-sample test, as in previous analysis of dynamic spatial proteomics experiments [26]. This framework directly tests perturbations to the quantitative mass-spectrometry profiles and hence can detect subtle changes in the steady-state localization of proteins between conditions. The posterior probability was used to quantify the extent of the difference between the DMSO and VX-809 localization profiles and proteins with a posterior probability of difference equal to 1 were considered to have perturbed steady-state localization. With these parameters, we found 231 proteins with potentially perturbed subcellular localization. These proteins are listed in Appendix A, along with their spatial localization in each condition and the corresponding Bayes score. Furthermore, for 76 of these proteins, it was impossible to pinpoint the exact direction of the translocation event, as they remained unlabelled/unknown following SVM classification. Their ambiguous status can be attributed to missing the classification threshold and may represent proteins residing in multiple locations, for example, proteins involved in trafficking pathways. There are several proteins classified as movers but predicted to localize within the same organelle (*ER* to *ER*, *PM* to *PM*…), thus suggesting small perturbations to their localization dynamics. Finally, other proteins were classified as moving from and to unknown locations. For example, our LOPIT-DC maps show a stark differential localization of Cathepsin L2 moving from “unknown” to “unknown.” This protein does not lie within the set threshold of 5% FDR for SVM classification but appears to change from a more lysosomal (SVM = 0.489) to a more ER (SVM = 0.72) location. The two other cathepsin proteins detected in our data (cathepsin B and D) stay in the lysosomes (Appendix A). The alluvial plot in Appendix A summarizes the changes in subcellular localization of the proteins that were identified to have differentially localized upon treatment with VX-809.

### 2.5. CFTR Expression and Localization

The LOPIT-DC experiment did not highlight CFTR among the proteins changing their localization upon treatment. This result was not surprising: F508del-CFTR resides mainly in the ER, as opposed to the wt-CFTR, mainly located in the PM [27]. Moreover, CFBE41o- cells stably overexpress *only* F508del-CFTR, whose primary localization is the ER [27]. LOPIT-DC provides an average picture of the primary localization of each protein across all cells in the sample, i.e., across all cellular states [28] and it correctly assigned ER as the main subcellular localization of F508del-CFTR in both control DMSO and VX-809 treated conditions. Furthermore, CFTR levels of expression are not significantly increased by VX-809, as demonstrated by our SWATH proteomics data (Appendix A). Additionally, VX-809 provides only a limited rescue of F508del-CFTR. This was demonstrated using western blotting, immunolocalization, and yellow fluorescent protein (YFP)-based assays, which showed only partial rescue of CFTR maturation, PM localization of CFTR and CFTR activity, respectively, in response to VX-809 treatment (Figure 3A–C). The partial efficacy of VX-809 as a modulator of CFTR trafficking was also confirmed also in primary bronchial epithelia, as evidenced by short-circuit current recordings (Figure 3D). This minor quantitative change in location is difficult to capture by LOPIT-DC.

Based on both expression and LOPIT-DC data, it is evident that treatment with VX-809 does not markedly alter both protein abundance and localization of F508del-CFTR. VX-809 treatment. However, VX-809 did appear to affect the localization of other proteins: 45 proteins did translocate between different organelles or reach a well-defined cell localization moving from multiple locations, as depicted in Appendix A. Among these moving proteins, 25 proteins were also quantified in the SWATH expression experiment (Appendix A) and showed no significant alteration in protein expression upon VX-809 treatment. This confirms that LOPIT-DC highlighted pure translocation events rather than changes in protein expression for this subset of proteins. Interestingly, of the 45 translocating proteins, 24 trafficked to and from the mitochondria and 13 of these (reported in Table 1) did not have a well-defined localization in the control condition, but their steady-state localization was classified as mitochondrial after VX-809 treatment. Again, for 6 of these 13, the absence of significant changes in expression was confirmed by the SWATH dataset.

We found that the dataset of differentially localized proteins (Appendix A) was highly enriched in mitochondrial components and redox functions, as summarized in Figure 4. The full set of results of network analysis of the translocators is reported in Appendix A.

Indeed, mitochondrial involvement in CF has been explored extensively in the past [31]. Before the cloning of CFTR, which allowed to demonstrate its role as a PManion channel, CF was thought to be a mitochondrial disease [32] given the observed impairment of cellular energetics, the accumulation of reactive oxygen species (ROS) and the increased glucose excretion in the air–liquid surface [33]. Impairment of mitochondrial functions, such as oxidative phosphorylation, has been demonstrated to occur in CF cells [34]. Surprisingly, our findings now reveal that pharmacological rescue of mutant CFTR protein is paralleled by the spatial rearrangement of mitochondrial proteins.

### 2.6. Mitochondrial and Peroxisomal Reorganization

The mitochondrial network is dynamic, with conformations that vary between a tubular continuum and a fragmented state [35]. Network morphology has been linked with the energy state in different cell types [35,36]. The mitochondrial morphology is also determined by the organelle motility, given that mitochondria are transported along microtubules utilizing ATP-dependent motor proteins [35]. In particular, it has been demonstrated that mitochondrial morphology changes due to bioenergetic stress [35]. Based on the results we obtained for the F508del-CFTR CFBE41o- subcellular proteome (and the lack of changes in expression of these same proteins), we hypothesized that mitochondria may undergo a structural rearrangement upon treatment with VX-809. To verify our hypothesis, we undertook an extensive analysis of mitochondrial network morphology in CFBE41o- cells using high-content imaging and analysis. To this aim, we utilized three different CFBE41o- cell lines: F508del-CFTR CFBE41o-, wildtype-CFTR (wt-CFTR) CFBE41o- and parental CFBE41o-. This was to monitor (1) changes possibly related to mutant CFTR expression and/or rescue, (2) changes possibly related to CFTR expression but presumably unrelated to CFTR rescue, and (3) possible drug effects unrelated to CFTR rescue, respectively. For these cells, two mitochondrial markers were used, TOMM20 and MitoTracker Red dye, to assess mitochondrial network morphology and mitochondrial membrane potential, respectively. In F508del-CFTR CFBE41o- cells, but not in wt-CFTR-expressing or parental CFBE41o- cells, under control conditions, TOMM20 staining showed mitochondria are compact around the perinuclear region (Figure 5A). Upon treatment with VX-809, mitochondria undergo a significant spatial rearrangement within the cell cytoplasm and appear to be more dispersed (Figure 5A,B). This compact, perinuclear morphology of the mitochondria resembled the mitochondrial network changes occurring following treatment with CCCP, a de-coupler of mitochondrial oxidative phosphorylation [35]. In our case, this “stressed” phenotype appears to resemble our control (DMSO-treated F508del-CFTR expressing cells). Following treatment with VX-809, F508del-CFTR CFBE41o- cells, but not wt-CFTR-expressing or parental CFBE41o- cells, had increased total signal of MitoTracker Red incorporated into mitochondria as compared to control, suggesting increased mitochondria viability (Figure 5C,D). On the contrary, the total signal of TOMM20 was unchanged (Figure 5B).

Our data indicate that, besides morphology and structure, mitochondrial functionality was also ameliorated by VX-809, as demonstrated by the increased MitoTracker uptake. Moreover, our data on parental and wt-CFTR expressing CFBE41o- cells indicates that this effect was likely specifically associated with the rescue of CFTR and it was not due to off-target (non-CFTR-related) effects. Another observation within the LOPIT-DC data was the differential localization of PEX13, moving from undefined locations to peroxisomes upon VX-809 treatment (Appendix A). PEX proteins are needed for peroxisome biogenesis: mutations in PEX genes can lead to lethal neurometabolic disorders of the Zellweger syndrome spectrum [37]. In particular, PEX13 homo-oligomers are important for the import of peroxisomal matrix proteins [37] during peroxisome biogenesis. Cells bearing ZSS-associated mutations in PEX13 display defective autophagy of damaged mitochondria mitophagy [38]. Interestingly, when challenged with CCCP, PEX13-deficient cells do not display the distinctive compaction of the mitochondrial network around the perinuclear region and have impaired clearance of damaged mitochondria by PEX13-mediated mitophagy [38]. While PEX13 is not represented in our SWATH dataset, and we thus have no direct data on its expression, taking into account the observed relocalization of PEX13, we investigated peroxisomal distribution in CFBE41o- cells expressing either mutant or wt-CFTR, as well as parental CFBE41o- cells, under resting condition and following VX-809 treatment (Figure 6). Following treatment with VX-809, in F508del-CFTR cells, the total number of peroxisomes per cell (i.e., PMP70-positive peroxisomes, Figure 6B, left graph was reduced from approximately 160 to 140 total peroxisomes/cell. No significant reduction was found in parental or wt-CFTR CFBE41o- cells. Notably, the number of PEX13-positive peroxisomes is severely and significantly reduced upon treatment with VX-809 from approximately 110 to 65 PEX13-positive peroxisomes/cell in cells expressing mutant CFTR. But no significant change in PEX13 positive peroxisomes was found in parental or wt-CFTR CFBE41o- cells (Figure 6B, right graph). With the combined evidence from our LOPIT-DC data, network enrichment analysis and mitochondrial and peroxisomal imaging assays, we hypothesize that the increased expression of PEX13 in F508del-CFTR cells helps the removal of “stressed” mitochondria. Therefore, when ameliorating mitochondrial functionality using VX-809, the peroxisomal network is also modulatedAs for the mitochondria rearrangement, the absence of any effect in parental and wt-CFTR expressing cells indicates that this effect is likely specifically related to CFTR rescue by VX-809.

As a further step, we aimed to validate our findings by translating these results to primary bronchial cells, obtained from F508del/F508del CF subjects. Well-differentiated bronchial epithelia from two F508del/F508del CF patients and, for comparison, from two non-CF subjects were grown under air–liquid interface for 20 days, and then treated for 24 h with DMSO alone or with VX-809 (3 µM). Epithelia were subsequently fixed and stained for the mitochondrial marker TOMM20 (Figure 7A) or the peroxisomal markers PMP-70 and PEX13 (Figure 7C). Differences in the mitochondrial network were investigated by performing TOMM20 signal texture analysis. As calculated by SER (Spots, Edges, Ridges; see Methods) features, TOMM20 signal was significantly different in CF, with the presence of very dense or scattered regions, vs. non-CF epithelial cells, having a more uniform texture (Figure 7B). Compared to DMSO alone, the treatment with VX-809 altered TOMM20 signal only in F508del epithelia, with a decrease in signal irregularities, with SER features values more similar to those of non-CF epithelia (Figure 7B). We then investigated peroxisomal distribution in epithelial cells by comparing PMP-70 and PEX13 staining (Figure 7C). Interestingly, while the percentage of PEX13-positive peroxisomes was approximately 50% in non-CF epithelia (either DMSO- or VX-809-treated), we observed concomitant PMP-70 and PEX13 staining in >90% of the peroxisomes in CF epithelia. Treatment with VX-809 markedly decreases the percentage of PEX13 peroxisomes to approximately 70–75% compared to DMSO (Figure 7D).

## 3. Discussion

In this work, we performed an extensive characterization of the changes in the cell proteome of F508del-CFTR-expressing CFBE41o- cells after the treatment with VX-809 (Lumacaftor), a drug that helps CFTR trafficking at the PM by correcting its aberrant folding. We first explored the alterations in protein expression by assessing the relative abundance of 3432 proteins. Our results indicated that a 24-h exposure to this drug does not significantly alter the protein expression of these proteins. We then captured the first “cell-wide” subcellular proteome of F508del-CFTR-expressing CFBE41o- using LOPIT-DC, obtaining information for over 4800 proteins. This spatial map provides a valuable resource for the CF research community. Additionally, using this method, we also investigated changes in the subcellular proteome of F508del-CFTR-expressing CFBE41o- cells following VX-809 treatment. We observed that no dramatic alterations to the global protein localization occur in this cell model following 24-h exposure to VX-809, but we recorded some subtle, yet very important, changes in protein location. This clearly demonstrated that LOPIT-DC is sufficiently reproducible and sensitive enough to detect subtle changes in location. Its application in this system revealed the relocalization of a set of proteins trafficking to and from the mitochondria upon CFTR rescue. Within this set of proteins, 13 have not previously been associated with CF or CFTR rescue. The role of mitochondria in CF has been previously investigated [32,35] and still attracts a great deal of attention in CF research. Recent literature [39] points to an active role of CF mitochondria in sustaining the inflammatory response. Prompted by the results of our “cell-wide” proteomics survey, we then specifically investigated mitochondria morphology, function and distribution in F508del-CFTR-expressing CFBE41o- model pre- and post-treatment with VX-809 by high-content confocal microscopy. Our data demonstrate that, upon VX-809 treatment, mitochondria and peroxisomes change their localization and morphology within the cell. Our experiments on parental cells and cells expressing wt-CFTR also indicate that these effects are specifically related to the rescue of CFTR triggered by VX-809. More interestingly, these findings were confirmed in primary epithelia derived from F508del homozygous patients and non-CF subjects. The association of VX-809 with significant improvement of mitochondrial functionality, such as oxygen consumption, has been previously demonstrated [40]. Our results support this observation, adding indeed a detailed insight into the specific molecular components involved in this rearrangement. In this respect, the observed relocalization of cathepsin L2 is intriguing. The inhibition of target proteases has already been proposed for CF [41], but this molecule has never specifically been associated with CF before, although it has been proposed as a target for pulmonary fibrosis [42]. Given the involvement of Cathepsin L2 in inflammation [43] and its role in preventing mitochondrial redox stress [44], we could hypothesize that its relocalization is broadly related to the amelioration of the CF phenotype induced by CFTR rescue. We collected expression data on approximately half of the proteome mapped by LOPIT-DC: we do not have evidence to support significant changes in the overall protein expression profiles following VX-809. We can thus infer that most of the alterations highlighted by our spatial proteomics experiment are related to bona fide protein translocation events, rather than to significant changes in their expression. It should be noted, however, that small changes in protein expression, especially of lower abundant proteins may be difficult to measure by the methods employed in this study. Moreover, we cannot rule out that observed protein relocalization could be a function of copies of a specific protein in one location being degraded and newly synthesized copies of the same proteins trafficked to an alternative location, leading to no concomitant change in the overall abundance of the protein. To disentangle translocation of existing proteins from differential synthesis and degradation, future experiments might envisage the use of in vivo isotopic labelling protocols, for example, pulsed-SILAC [45,46], a technique that allows to specifically –determine de novo protein synthesis over short periods of times. Despite these limitations, the present work and the data we acquired represent a useful resource in the CF field. We believe that this exemplifies a new way of looking at the role of proteins in CF, i.e., by extending investigation beyond protein expression profiling, by assessing the importance of their spatial localization within the cell. This new perspective will aid research into the mechanisms of pharmacological intervention for this debilitating disease.

## 4. Materials and Methods

### 4.1. Chemicals, Reagents and Instruments

All the chemicals and reagents were purchased from Merck (Darmstadt, Germany) unless otherwise indicated. All the reagents for cell culture were purchased from Euroclone (Milano, Italy). VX-809 was purchased from Biosynth Carbosynth (Berkshire, UK). VX-809 was used (final concentration) at 1 µM for proteomics and 3 µM for validation studies. Considering the dose–response relationship for VX-809 as mutant CFTR corrector, both the concentrations results in maximal rescue without any toxic effect [47].

### 4.2. CFBE41o- Cell Culture

CFBE41o- cells transfected to stably express F508del-CFTR and the halide-sensitive yellow fluorescent protein (HS-YFP) were cultured as already described [48,49]. Briefly, the cells were cultured until confluence in MEM medium (Euroclone) supplemented with 10% fetal calf serum (Euroclone) and 1% l-glutamine (Euroclone). The medium was also supplemented with 0.75 mg/mL G418 (Invivogen, San Diego, CA, USA) and 2 µg/mL Puromycin (Invivogen) as resistance markers for F508del-CFTR and HS-YFP, respectively.

### 4.3. Expression Proteomics: Sample Preparation and SWATH Data Acquisition

CFBE41o- cells overexpressing F508del-CFTR (5 million cells per condition) were plated on 60-millimeter Petri and treated for further 24 h with vehicle alone (DMSO 0.1%), or VX-809 1 μM. Five independent biological replicates were prepared. Cells were harvested with trypsin, washed in PBS Ca^2+^/Mg^2+^ free and pelleted by centrifugation 4 °C at 300× *g*, the supernatant was discarded. Cells pellets were then re-suspended in H_2_O (50 μL), transferred to glass vials, and crushed with 1 mL isopropanol, vortexed for 10 min and sonicated for 10 min at room temperature (RT). Samples were then centrifuged at 20,000× *g* for 20 min, and, after discarding the supernatant, the pellets were dried under nitrogen under the fume hood. The protein lysates were then re-dissolved in 400 µL of SDS 2% and a BCA assay (Pierce ThermoFisher, Waltham, MA, USA 02451) was performed to measure protein concentration. From each sample, a volume corresponding to 50 µg of proteins was transferred to Eppendorf tubes and disulphide bonds were reduced with 10 µL of 100 mM dithiothreitol (in 50 mM NH_4_HCO_3_, pH 8) for 30 min at 56 °C. For the alkylation of cysteine residues, 30 µL of 100 mM iodoacetamide (in 50 mM NH_4_HCO_3_, pH 8) were added and the samples were incubated at RT for 20 min in the dark. Proteins were precipitated in acetone overnight at −20 °C. Samples were then centrifuged at 20,000× *g* for 30 min at 4 °C, the supernatant was removed and pellets were washed with 500 µL of cold methanol, vortexed for 1 min and centrifuged at 20,000× *g* for 30 min at 4 °C. The supernatant was removed and protein pellets were dried again under the fume hood. Each sample was then re-dissolved in 70 µL of RapiGest solution (Waters Inc., Milford, MS, USA). After vortexing for 1 h at RT, 2 µL of trypsin (0.5 µg/µL) were added. Tubes were incubated at 37 °C in a shaker at 600 rpm overnight. To remove RapiGest before analysis, 7 µL of trifluoroacetic acid 5% were added and samples were incubated at 37 °C in a shaker at 600 rpm for 45 min. Samples were then centrifuged at 20,000× *g* for 30 min at 4 °C. The supernatant was collected, transferred to new tubes and dried down in a speed-vac evaporator. The day of analysis, the peptides were re-suspended in 50 µL of 3% acetonitrile added with 0.1% formic acid. QC samples were prepared by pooling together equal aliquots of the ten samples (5 for control DMSO and 5 for VX-809). Samples from controls, treated and QC groups were then acquired in a randomized way by high-resolution LC-MS for a total of 15 runs. A nanoUPLC Acquity chromatographic system (Waters, Milford, MA, USA) coupled to a TripleTof 5600+ mass spectrometer (SCIEX) was used. Eluents were A (water + 0.1% formic acid) and B (acetonitrile + 0.1% formic acid). Injection volume was set to 5 µL (Full Loop) and flow rate was set to 5 μL/min. The peptides were loaded and desalted on a trapping column (Guard column YMC-Triart C18, 3 µm particle size, 0.5 mm× 5 mm, 1/32″) at 1% acetonitrile + 0.1% formic acid for 5 min then moved on a reversed-phase C18 column (Eksigent C18, 3 µm particle size, 0.3 mm× 150 mm format), kept at 45 °C. Peptides were eluted with the following gradient program: 0.0–1.0 min 2% B; 1.0–60.0 min 2 to 35% B; 60.0–63.0 min 35 to 95% B; 63.0–68.0 min 95% B and 68.0–68.1 min 95 back to 2% B. The column was then reconditioned for 12 min. The total run time was 80 min. A mixture of tryptic peptides from bovine serum albumin was analyzed every 5 runs and used for mass recalibration. Analysis was performed in SWATH DIA (data-independent acquisitions) mode, in positive ESI mode. A TOF MS Scan was set from 350 to 1250 *m*/*z* with a survey scan of 50 ms. 100 SWATH experiments, with an accumulation time of 25 ms each, were collected from 100 to 1500 *m*/*z* in High Sensitivity mode. The total cycle time was 2.6 s. The collision energy for each SWATH experiment was automatically calculated by the acquisition software using the equation: CE = 0.063 (*m*/*z*)–3.24. The ion source parameters were ion spray voltage floating at 5000 V, Ion source gas 1 at 30, curtain gas at 30 and declustering potential at 80 V.

### 4.4. SWATH Data Analysis

The SWATH-MS raw data were analyzed using the SWATH Acquisition MicroApp 2.0.1.2133 incorporated in PeakView software 2.2 (SCIEX). An ion library specifically dedicated to CFBE41o- cells was used [22]. For protein quantification, the following peptide filter parameters were used: number of peptides per protein at 6, number of transitions per peptide at 6, peptide confidence threshold at 99%, FDR threshold at 1%, maximum mass tolerance at 50 ppm and modified peptides were excluded. For further processing, only proteins quantified by at least 2 peptides were retained. Data were normalized using the MLR algorithm [50]. The following data analysis steps were performed using MetaboAnalyst web-based software [51]. Only features showing a percent coefficient of variation (CV%) lower than 25% in QC samples were retained, for a total of 3432 quantified proteins (Appendix A). Proteins quantified but showing higher CV% in quality check samples (pools of aliquots from all the samples) were excluded. Data were not transformed but simply Pareto-scaled to perform PCA analysis and the subsequent *t*-test statistics.

### 4.5. LOPIT-DC

#### Cell Culture and VX-809 Treatment

CFBE41o- cells stably overexpressing F508del-CFTR were selected and cultured until confluent. For LOPIT-DC experiment, 70 million cells per sample are needed. Three independent biological replicates were prepared, where experimental conditions for each replicate were processed simultaneously. For each replicate, cells were incubated for 24 h with MEM media containing either the VX-809 1 µM or the vehicle (DMSO, used as control).

### 4.6. Cell Lysis and Subcellular Fractionation

Cells were harvested with TrypLE Express Enzyme (Gibco) and pelleted at 300× *g* for 7 min at 4 °C. Cell pellets were suspended in 5 mL of isotonic lysis buffer (0.25 M sucrose, 10 mM HEPES pH 7.4, 2 mM EDTA, 2 mM magnesium acetate, protease inhibitors). Cells were then lysed using a pair of 1-millilitre syringes in a ball-bearing homogenizer (Isobiotec): each aliquot of cell suspension was passed through the homogenizer chamber 30 times using an 18-micrometre ball-bearing clearance, keeping the homogenizer on ice. Cell lysates were processed following the differential centrifugation workflow already described (21).

### 4.7. Evaluation of Organelles Separation through Western-Blot (WB)

We tested the efficiency of organelle separation via differential centrifugation with western blot analysis, using a set of protein markers. After performing protein quantitation using a BCA assay (Pierce ThermoFisher, Waltham, MA, USA 02451), 2 µg of proteins from each fraction was separated onto gradient 4–20% Mini-PROTEAN TGX Precast gel (Bio-rad laboratories Inc., Hercules, CA, USA) and then transferred onto a Polyvinylidene fluoride (PVDF) membrane (Bio-rad laboratories Inc.). The membranes were blocked with 5% milk dissolved in TBS-T and washed three times with TBS-T; membranes were cut into strips at the molecular weight of the corresponding primary antibodies (see Table 2). Sufficient organelle separation is achieved when all the organelle-specific marker proteins show a unique and distinct distribution profile across the LOPIT-DC fractions. Appendix A shows the resulting Western Blot and the optimized organelle separation obtained for CFBE41o- cells.

After the incubation at room temperature for 1 h with HRP-linked anti-rabbit secondary antibody (dilution 1:10,000), Amersham ECL Prime Western Blotting Detection Reagent (GE Healthcare) was added to the membranes and the signal was detected using X-ray films.

### 4.8. Protein Digestion and TMT Labelling

50 µg of proteins from each organellar fraction was transferred into new Eppendorf Lo-bind tubes and in-solution digestion was performed. Disulfide bonds were reduced with 10 µL dithiothreitol (DTT) 100 mM at 56 °C for 30 min; the corresponding cysteine residues were then alkylated with 30 µL of iodoacetamide (IAA) 100 mM for 20 min in the dark. Proteins were precipitated overnight with cold acetone at −20 °C. After a centrifugation step (20,000× *g* for 30 min at 4 °C), the supernatant was discarded. The pellet was then dissolved for overnight trypsin digestion at 37 °C. The tryptic peptides were labelled with TMT isobaric tagging reagents (Thermo Fisher Scientific, Waltham, MA, USA 02451) according to the manufacturer’s protocol. Since each TMT kit can be used to tag up to 100 µg of proteins, half of the TMT is used for each sample (50 µg of proteins). The TMT tags were equilibrated at room temperature and each tag was dissolved in 82 µL of acetonitrile (ACN). After vortexing and centrifugation for 2 min at 2000× *g*, 41 µL of TMT tag solution was added to label each fraction, following the scheme shown in Appendix A, for two hours on a shaker at room temperature. The labelling reaction was quenched by adding 8 µL of 5% (*w*/*v*) hydroxylamine and incubated for 45 min at room temperature on a shaker. Another quenching step was performed by incubation with 100 µL of MilliQ water for 30 min at room temperature on a shaker. After the TMT labelling, all the fractions were pooled together and dried using SpeedVac.

### 4.9. Peptide Fractionation and LC-MS/MS Analysis

Dried TMT-labelled peptides were resuspended in 1.8 mL of 0.1% TFA (trifluoroacetic acid) and half the solution was pre-fractionated using the Pierce High pH Reversed-Phase Peptide Fractionation Kit (Thermo Fisher Scientific) due to the binding capacity of the columns and according to manufacturer’s protocol. Only half of the volume was used because of the binding capacity of the peptide-fractionation columns. In summary, the columns were activated and equilibrated before loading the samples. The peptides were then washed and eluted with increasing concentrations of ACN (5–10–12.5–15–17.5–20–22.5 and 25%). To decrease the number of the samples for downstream MS analysis (8 fractions of peptides per sample, 6 samples in total), the resulting fractions were combined into 4 fractions per sample (Fraction 1 + 5; Fraction 2 + 6; Fraction 3 + 7; Fraction 4 + 8). All MS runs were performed on an Orbitrap Eclipse Tribrid Mass Spectrometer coupled to a Dionex Ultimate TM 3000 RSLnano system (Thermo Fisher Scientific). Each of the fractionated samples was dissolved in 60 µL of 0.1% formic acid (FA) and 2 µL were injected onto a micro precolumn (300-μm × 5 mm, C18 PepMap 100, 5-μm particle size, 100-Å pore size, Thermo Fisher Scientific) for the trapping phase (3 min). After the switching of the valve from load to inject, the peptides were separated on a 200 cm µPAC^TM^ column (PharmaFluidics, Ghent, Belgium). The nanoLC gradient was set as follows: from 3 to 40% of ACN in water + 0.1% FA at 300 nL/min for 340 min. The column was then washed using 95% of ACN + 0.1% FA for 9 min and re-equilibrated for 40 min. Total run time was 390 min. The MS workflow parameters were set as follows using the Method Editor in XCalibur (Thermo Fisher Scientific) for the SPS-MS3 acquisition method: Detector type: Orbitrap—Resolution: 120,000—Mass range: Normal—Use quadrupole isolation: Yes—Scan range: 400–1500—RF lens: 30%—AGC target: 40,000—Max inject time: Auto—Microscans: 1—Data type: Profile—Polarity: Positive—Monoisotopic peak determination: Peptide—Relax restrictions when too few precursors are found: Yes—Include charge state(s): 2–6—Exclude after n times: 1—Exclusion duration (s): 70—Mass tolerance (p.p.m.): Low: 10; high: 10—Exclude isotopes: Yes—Perform dependent scan on single charge state per precursor only: Yes—Intensity threshold: 5.0 × 10^3^—Data-dependent mode: Cycle time —Number of scan event types: 1—Scan event type 1: No condition—MSn level: 2—Isolation mode: Quadrupole—Isolation window (*m*/*z*): 0.7—Activation type: CID —CID collision energy (%): 35—Activation Q: 0.25—Detector type: Ion trap—Scan range mode: Auto—*m*/*z*: Normal—Ion trap scan rate: Rapid—AGC target: 1.0 × 10^4^—Max inject time (ms): 35—Microscans: 1—Data type: Centroid—Mass range: 400–1500—Exclusion mass width: *m*/*z*: Low: 0; high: 0—Reagent: TMT—Precursor priority: Most intense—Scan event type 1: No condition—MSn level: 3—Isolation mode: Quadrupole—Isolation window (*m*/*z*): 0.7—Activation type: HCD —Collision energy (%): 65—Detector type: Orbitrap—Scan range mode: Define—*m*/*z*: 100–500—AGC target: 1.0 × 10^4^—Max inject time (ms): 105—Microscans: 1—Data type: Centroid. An electrospray voltage of 2.1 kV was applied. The mass spectrometer was operated in positive ion data-dependent mode for SPS-MS3 with the real-time (RT) search using the *Homo sapiens* Swiss proteome (downloaded 09/04/2018 as FASTA file). The parameters for the RT-search were enzyme: trypsin; max variable mods: 2; max missed cleavage: 1; Static modification: carbamidomethylation on cysteine and TMT-10plex on lysine and peptide N terminus; variable modifications: oxidation of methionine.

### 4.10. LOPIT-DC Data Processing

Raw files were processed with Proteome Discoverer version v2.3 (Thermo Fisher Scientific). The acquired MS/MS spectra were searched using Proteome Discoverer with Mascot and SequestHF algorithm nodes against the *Homo sapiens* database (canonical and isoform, 42,118 sequences, downloaded on 04/11/2016) together with common contaminants (cRAP). The research parameters were set as follows: precursor mass tolerance: 10 ppm; fragment mass tolerance: 0.6 Da; enzyme used for digestion: Trypsin; maximum missed cleavage: 2; fixed modifications: carbamidomethylation of cysteine and TMT10plex tagging of lysine and peptide N terminus for TMT labelled samples; dynamic modifications: oxidation of methionine and deamidated asparagine and glutamine. Percolator node was used for false discovery rate estimation and only rank 1 peptide identifications of high confidence (FDR < 1%) were accepted. TMT reporter values were assessed through Proteome Discoverer v2.3 using the Most Confident Centroid method for peak integration and integration tolerance of 20 p.p.m. Reporter ion intensities were adjusted to correct for the isotopic impurities of the different TMT reagents (according to the manufacturer specifications for the respective batch number). Percolator was used to assess the false discovery rate (FDR) and only high-confidence peptides were retained. Additional data reduction filters were: peptide rank = 1 and ion score > 20. Quantification at the MS3 level was performed within the Proteome Discoverer workflow using the centroid sum method and an integration tolerance of 2 mmu. Isotope impurity correction factors were applied. Protein grouping was carried out according to the minimum parsimony principle and the median of all sum-normalised PSM ratios belonging to each protein group was calculated as the protein group quantitation value. Only proteins with a full reporter ion series were retained. In three individual biological replicates of CFBE41o- cultures treated with DMSO as control, we identified 5949 ± 168 proteins at 1% FDR; following a well-established protocol for the analysis of LOPIT datasets [25], each replicate was separately normalized by sum to 1 across the 10 channels for each replicate respectively. After the removal of the missing values, the three replicates were concatenated, leaving a total of 4842 proteins with spatial information common to the 3 replicates for the control samples. The quantified proteins were analyzed using the R Bioconductor [52] packages MSnbase (v. 2.12) [53] and pRoloc (v1.26) [25] following a well-established protocol for LOPIT datasets [25]. After the removal of the missing values, the three replicates were combined to obtain a map of the organelles for each condition. A list of 548 manually-curated marker proteins (defined as proteins known to localize only in one specific organelle) was used to pinpoint 11 subcellular localizations: cytosol, ER, proteasome, Golgi, lysosomes, mitochondria, nucleus, chromatin, ribosome, peroxisomes and plasma membrane. To predict protein localization of the proteins quantified both in the controls and the treated samples, a support vector machine (SVM) classifier was used, following a protocol already available in the literature [25]. The algorithmic performance was estimated by the use of 100 rounds of five-fold cross-validation (creating stratified training/testing partitions). Classifier accuracy was estimated using the macro F1 score (the harmonic mean of precision and recall). The best values for sigma (the inverse kernel width for the radial basis kernel) and cost (the cost of constraints violation) were 0.1 and 16, respectively for both the control and the treated samples. All the proteins were then ordered according to their SVM score and the proteins below a 5% FDR cutoff were classified as proteins with unknown localization. To ensure that an acceptable subcellular resolution was achieved for each replicate, we plot PCA maps for each replicate (Appendix A). Appendix A shows the combined LOPIT-DC distribution profiles across the subcellular fractions collected by differential centrifugation of proteins classified to 11 subcellular localizations using SVM classification, with a 5% FDR cut-off for both control and treated triplicates.

### 4.11. Statistical Analysis of Differential Localization

Following Shin et al. [26], to detect perturbations in the quantitative protein profiles, we applied a Bayesian non-parametric two-sample test. First, the data was transformed using the additive log ratio transform [54]. We then proceed to test whether the protein profiles were different between the control and treatment. Formally, we test against two contrasting models. The first model (null model) posits that the quantitative protein profiles in each experiment (control and treatment) were drawn from an identical shared distribution. Whilst the second supposes that there were independent models for each of the control and treatment. The log Bayes factor was used to objectively determine support for one model over the other, where larger log Bayes factors were considered support for the independent model [55]. A non-parametric prior over functions, the Gaussian process, was specified with squared exponential covariance. Default Gamma priors were used for the hyperparameters of the Gaussian process [56]. The natural logarithm of the Bayes factors is reported. Bayes factors were converted to posterior probabilities and posterior probabilities equal to 1 were considered evidence in favor of the independent model. We assumed a priori that the probability of the independent model was 0.01.

### 4.12. Enrichment Analysis

Gene enrichment and functional overrepresentation analysis were performed using FunRich software (version 3.1.3) [57]. The list of 231 proteins significantly changing their localization after VX-809 treatment was searched against the FunRich database which reports Homo sapiens data obtained from a set of publicly available resources, such as the Gene Ontology database and Uniprot.

### 4.13. YFP-Based Assay for CFTR Activity

For functional assays of CFTR activity, CFBE41o- cells co-expressing mutant CFTR and the Halide-Sensitive Yellow Fluoresce Protein (HS-YFP) were plated (50,000 cells/well) on clear-bottom 96-well black microplates (Corning Life Sciences). The following days, cells were treated with DMSO (vehicle alone) or VX-809 (3 µM) for 24 h. The microfluorimetric assay for the determination of CFTR activity based on the HS-YFP was described in detail in previous studies [49,58]. Briefly, prior to the assay, cells were incubated with PBS plus forskolin (20 µM) and VX-770 (1 µM) for 20 min at 37 °C to maximally stimulate F508del-CFTR. Cells were then transferred to a microplate reader (FluoStar Fluostar Optima; BMG Labtech, Offenburg, Germany) equipped with high-quality excitation (HQ500/20X: 500 ± 10 nm) and emission (HQ535/30M: 535 ± 15 nm) filters for YFP (Chroma Technology) for CFTR activity determination. YFP fluorescence was recorded for 12 s after the injection of an iodide-containing PBS (in which Cl- was replaced by I- to reach a final I- concentration of 100 mM). To determine I-influx rate, the final 11 s of the data for each well were fitted with an exponential function to extrapolate initial slope (dF/dt), after normalization to the initial background-subtracted fluorescence.

### 4.14. CFTR Localization

Analysis of CFTR localization was performed by immunofluorescence in formalin-fixed, wild-type- or F508del-CFTR expressing CFBE41o- cells under resting conditions or following 24 h treatment with VX-809 (3 µM) and, for comparison, in parental CFBE41o- cells, seeded on high-quality clear-bottom, 96-well black plates suitable for high-content imaging (CellCarrier, Perkin Elmer, Waltham, MA, USA). The protocol for CFTR immunostaining was previously described [59]. Mouse IgG1 anti-CFTR ab570 was used as the primary antibody (kindly provided by John Riordan, Ph.D., University of North Carolina—Chapel Hill, and Cystic Fibrosis Foundation) followed by an anti-mouse secondary antibody conjugated to AlexaFluor 488. Cell nuclei were counterstained with Hoechst 33342.

### 4.15. Western Blot for the Evaluation of VX-809 Corrector Activity

CFBE41o- cells were grown to confluence on 60-mm diameter dishes and lysed in RIPA buffer (50 mM Tris-HCl pH 7.4, 150 mM NaCl, 1% Triton X-100, 0.5% Sodium deoxycholate, 0.1% SDS) containing a complete protease inhibitor (Roche, Basel, Switzerland). Cell lysates were then processed as previously reported [29]. Equal amounts of protein were separated onto gradient (4–15%) Criterion TGX Precast gels (Bio-rad laboratories Inc., Hercules, CA, USA), transferred to nitrocellulose membrane with Trans-Blot Turbo system (Bio-rad Laboratories Inc.) and analyzed by Western blotting. CFTR and GAPDH proteins were detected using mouse monoclonal anti-CFTR (ab596, J.R. Riordan, University of North Carolina at Chapel Hill, and Cystic Fibrosis Foundation Therapeutics) and mouse monoclonal anti-GAPDH (sc-32233; Santa Cruz Biotechnology, Dallas, TX, USA) and horseradish peroxidase (HRP)-conjugated anti-mouse IgG (ab97023; Abcam, Cambridge, UK) and subsequently visualized by chemiluminescence using the SuperSignal West Femto Substrate (Thermo Scientific, Waltham, MA, USA). Chemiluminescence was monitored using the Molecular Imager ChemiDoc XRS System. Images were analyzed with ImageJ software (National Institutes of Health, Bethesda, MD, USA). Bands were analyzed as Region-Of-Interest (ROI) and normalized against the GAPDH loading control. Data are presented as mean ± SEM of independent experiments.

### 4.16. Culture and Differentiation of Primary Bronchial Epithelial Cells

The methods for the isolation, culture and differentiation of primary bronchial epithelial cells were previously reported [29]. In brief, epithelial cells were obtained from mainstem human bronchi, derived from individuals undergoing lung transplants. Epithelial cells were cultured in a serum-free medium (LHC9 mixed with RPMI 1640, 1:1) containing various hormones and supplements, favouring cell number expansion. To obtain differentiated epithelia, cells were seeded at high density on porous membranes (Snapwell inserts, Corning, Corning, NY, USA, code 3801). After 24 h, the serum-free medium was removed from both sides and, on the basolateral side only, replaced with Pneumacult ALI medium (StemCell Technologies, Vancouver, BC, Canada) and differentiation of cells (up to 16–18 days) was performed in air-liquid interface (ALI) condition.

The collection of bronchial epithelial cells (supported by Fondazione per la Ricerca sulla Fibrosi Cistica through the “Servizio Colture Primarie”) and their study to investigate the mechanisms of transepithelial ion transport were specifically approved by the Ethics Committee of the Istituto Giannina Gaslini following the guidelines of the Italian Ministry of Health (registration number: ANTECER, 042-09/07/2018). Each patient provided informed consent to the study using a form that was also approved by the Ethics Committee.

### 4.17. Analysis of CFTR-Mediated Transepithelial Ion Transport in Primary Bronchial Epithelial Cells

Snapwell inserts carrying differentiated bronchial epithelia were mounted in a vertical diffusion chamber resembling a Ussing chamber with internal fluid circulation. Apical and basolateral hemichambers were filled with a symmetrical solution containing: 126 mM NaCl, 0.38 mM KH_2_PO_4_, 2.13 mM K_2_HPO_4_, 1 mM MgSO_4_, 1 mM CaCl_2_, 24 mM NaHCO_3_, and 10 mM glucose, continuously bubbled with 5% CO_2_–95% air mixture. The temperature of the solution was kept at 37 °C. The transepithelial voltage was short-circuited with a voltage-clamp (DVC-1000, World Precision Instruments; VCC MC8 Physiologic Instruments) connected to the apical and basolateral chambers via Ag/AgCl electrodes and agar bridges (1 M KCl in 1% agar). The offset between voltage electrodes and the fluid resistance was adjusted to compensate for parameters before experiments. The short-circuit current was recorded by analogical to digital conversion on a personal computer.

To evaluate CFTR-mediated chloride secretion, DMSO- or VX-809 treated epithelia were exposed sequentially to different pharmacological agents to modulate specific ion channels and transporters. First, amiloride (10 µM) was added to block Na^+^ current mediated by the epithelial sodium channel (ENaC). Then, CFTR activity was elicited by applying the non-hydrolysable, membrane-permeable cAMP analogue CPT-cAMP (100 µM) followed by VX-770 (1 µM) to maximally activate CFTR. Finally, the selective CFTRinh-172 (inh-72, 10 µM; [60]) was added to inhibit CFTR-mediated currents. For each epithelium, the amplitude of the current drop caused by inh-172 was used as an estimate of total CFTR activity.

### 4.18. Analysis of Mitochondria Network and Peroxisomal Distribution

Mitochondrial network and peroxisomes were monitored by immunofluorescence in formalin-fixed, CFBE41o- cells or primary bronchial epithelia under resting conditions or following 24 h treatment with VX-809 (3 µM). Primary antibodies used for immunofluorescence staining include the following: mouse anti-TOMM20 (Santa Cruz, sc-17764), mouse anti-PEX13 (Santa Cruz, sc-271477), and rabbit anti-PMP70 (Thermo Fisher Scientific, PA1650). Secondary antibodies were conjugated to AlexaFluor 488 or AlexaFluor 647 (Thermo Fisher Scientific). Cell nuclei were counterstained with Hoechst 33342. High-content imaging was performed using an Opera Phenix (PerkinElmer) high-content screening system. Wells were imaged in confocal mode, using a 40X water-immersion objective. AlexaFluor 488 signal was laser-excited at 488 nm and the emission wavelengths were collected between 500 and 550 nm. Excitation and emission wavelengths for visualization of AlexaFluor 647 signal were 640 and between 650 and 760 nm, respectively. Excitation and emission wavelengths for visualization of Hoechst 33342 signal were 405 and between 435 and 480 nm, respectively. Image analysis of signal intensity and morphology, as well as automatic detection of signal spots (resembling peroxisomes), were performed using the Harmony software (version 4.9) of the Opera Phenix high-content system. The segmentation of cell cytoplasm into two regions, one comprising the perinuclear area (inner cytoplasmic region) and the second one comprising the more peripheral zone (outer cytoplasmic region) was performed by means of an automated algorithm, developed using machine-learning techniques. Analysis of the signal texture was performed using the PhenoLogic machine-learning algorithm of the Harmony software (version 4.9) of the Opera Phenix high-content system. Analysis of signal texture was based on evaluation of SER (Spots, Edges, Ridges) features, developed by PerkinElmer and included in the Harmony software of Opera Phenix.

## Figures and Tables

**Figure 1 cells-11-01938-f001:**
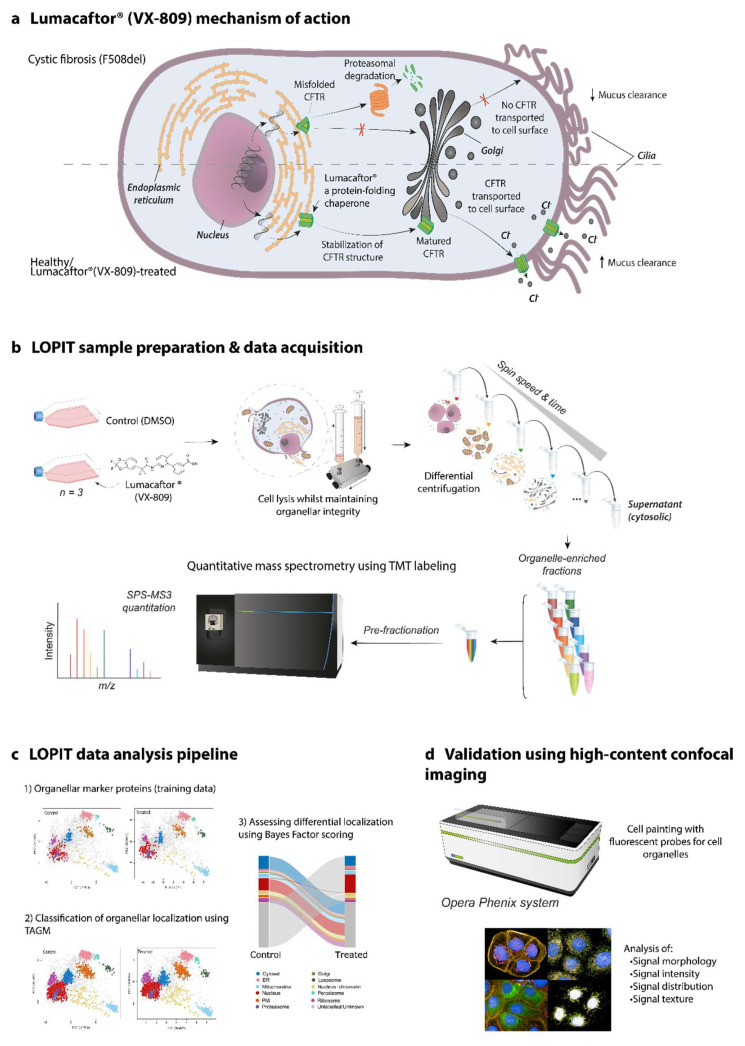
LOPIT-DC workflow applied to F508del-CFTR CFBE41o- cells. (**a**) Mechanism of action of Lumacaftor (VX-809), a drug used to promote the folding and trafficking of F508del-CFTR to the PM. (**b**) Schematic of LOPIT sample and data acquisition. After incubation with either VX-809 or control DMSO, cells were lysed and organelles separated by differential centrifugation into 10 fractions. Following tryptic digestion, the peptide content of each fraction was then labelled with a different isobaric tag (TMT), and fractions pooled together and analyzed by nanoLC-MS/MS. In each peptide MS/MS spectrum, the relative intensity of the TMT reporter ions is associated with the abundance of the corresponding protein in each fraction. (**c**) LOPIT data analysis pipeline. Based on the data from markers of known subcellular localization using multivariate data analysis, the spatial map of the CFBE41o- proteome was produced for each condition. Using SVM classification and Bayes factor scoring, changes in the subcellular localization of proteins following VX-809 treatment are assessed. (**d**) Validation of LOPIT-DC hits using high-content confocal imaging.

**Figure 2 cells-11-01938-f002:**
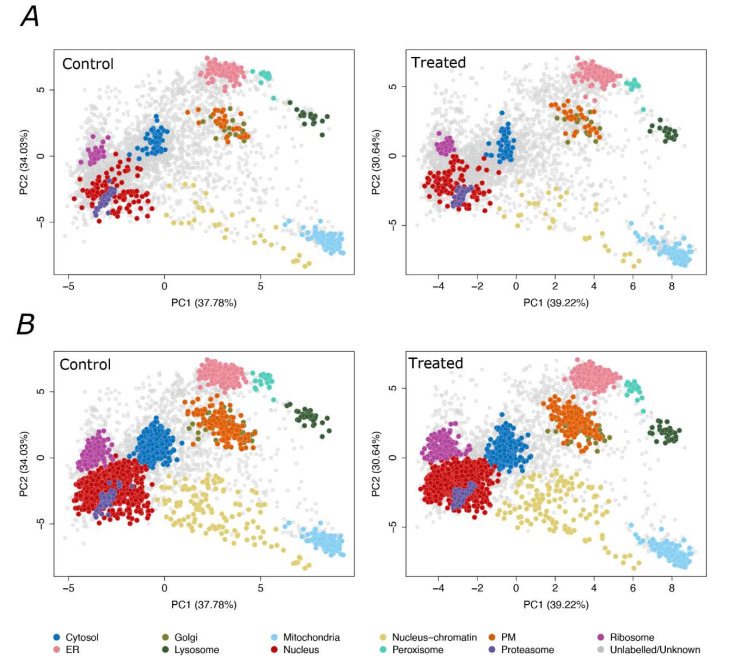
Subcellular maps of the F508del-CFTR CFBE41o- proteome. Each point corresponds to a single identified protein. (**A**) Principal component analysis (PCA) projections with subcellular marker proteins highlighted for 11 subcellular compartments (highlighted by different colours) in the control DMSO (left) and treated with VX-809 (right) experiments. (**B**) PCA plot of the Support Vector Machine (SVM) classification results after applying a 5% FDR cutoff; the proteins below the 5% FDR cutoff were classified as proteins with unlabeled/unknown localization.

**Figure 3 cells-11-01938-f003:**
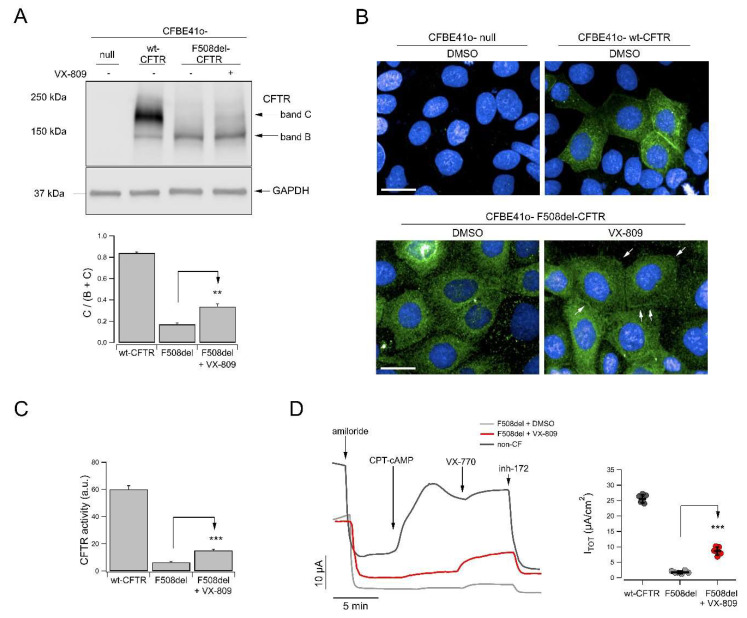
Evaluation of VX-809 corrector activity in human bronchial epithelial cells. (**A**) Biochemical analysis of CFTR expression pattern in total lysates from wild-type or F508del-CFTR expressing CFBE41o- cells after 24 h treatment (for mutant CFTR only) with vehicle alone (DMSO) or VX-809 (3 µM) prior to lysis. For comparison, whole lysates derived from CFBE41o- cells not expressing CFTR (null cells) are also shown as controls for antibody specificity. Arrows indicate mature, complex-glycosylated (band C) and immature, core-glycosylated (band B) forms of CFTR protein. The bar graph shows quantification of CFTR bands. Data are expressed as C band/B band ratio normalized for the value observed in cells treated with DMSO. Data are expressed as means ± SEM, *n* = 3 independent experiments. Statistical significance of VX-809 treatment was tested by Student’s *t*-test. Symbols indicate statistical significance versus DMSO: **, *p* < 0.01. (**B**) Representative confocal microphotographs showing F508del-CFTR or wild-type CFTR subcellular localization in CFBE41o- cells by immunofluorescence. Cells expressing mutant CFTR were incubated with vehicle (DMSO) or VX-809 (3 µM) for 24 h prior to fixation. CFTR was immunodetected using a mouse IgG1 anti-CFTR (ab570) as primary antibody followed by an anti-mouse secondary antibody conjugated to AlexaFluor 488. Cell nuclei were counterstained with Hoechst 33342. In wt-CFTR expressing cells, CFTR localization displays a membrane pattern. In DMSO-treated F508del-CFTR cells, CFTR protein resides in the perinuclear region. After treatment with corrector VX-809, some CFTR expression was apparent at the plasma membrane, although most of the protein was still localized in the perinuclear region. Scale bar: 50 µm. (**C**) YFP functional evaluation of CFTR channel activity on CFBE41o- cells expressing wild-type or F508del-CFTR and the halide-sensitive yellow fluorescent protein (HS-YFP), following 24 h treatment with vehicle alone (DMSO) or VX-809 (3 µM). Activity was measured after acute stimulation with the cAMP agonist forskolin (20 µM; [29,30]) plus CFTR potentiator VX-770 (1 µM). Symbols indicate statistical significance versus DMSO: ***, *p* < 0.001. (**D)** Representative recordings showing transepithelial ion transport in primary bronchial epithelial cells and the effect of 24 h cell treatment with vehicle alone (DMSO) or VX-809 (3 µM). Experiments were performed on epithelia derived from one F508del/F508del CF patient (ID donor: BE93) or, for comparison, from one non-CF subject (ID donor: BE177), with the short-circuit current technique. After inhibition of the epithelial sodium channel ENaC with amiloride (10 µM), cells were stimulated with the membrane-permeable cAMP analogue CPT-cAMP (100 µM) followed by VX-770 (1 µM) to maximally activate F508del-CFTR. Then, CFTR currents were blocked adding the CFTR inhibitor-172 (inh-72, 10 µM). The dot plot shows the total current sensitive to inh-172 (I_TOT_) measured in each epithelium, used as parameter of CFTR function. Asterisks indicate statistical significance versus DMSO: ***, *p* < 0.001. Only statistically significant comparisons are reported.

**Figure 4 cells-11-01938-f004:**
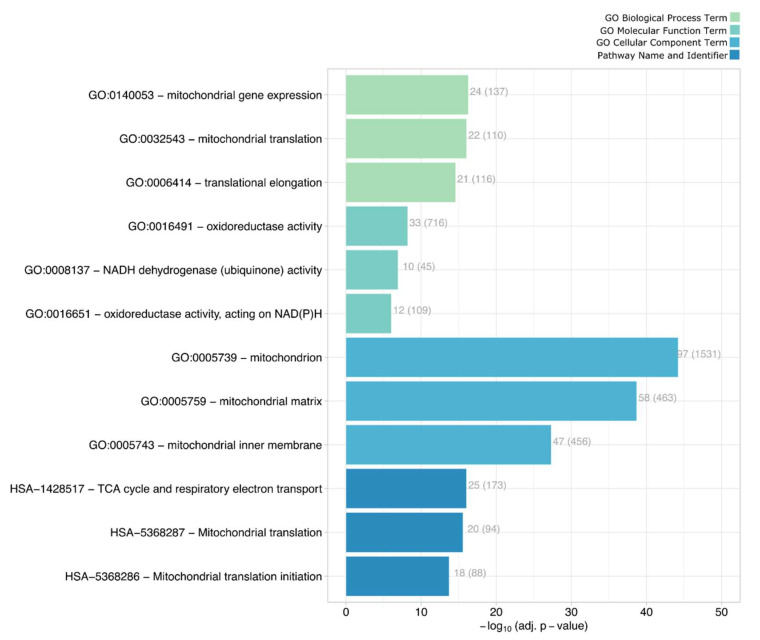
Most enriched processes, functions, components and pathways represented by the 231 movers. *X*-axis reports the corresponding negative log p-values, adjusted for multiple testings. The numbers at the end of the bars refer to the number of mapped proteins over the total number of proteins involved in the corresponding term (indicated in brackets).

**Figure 5 cells-11-01938-f005:**
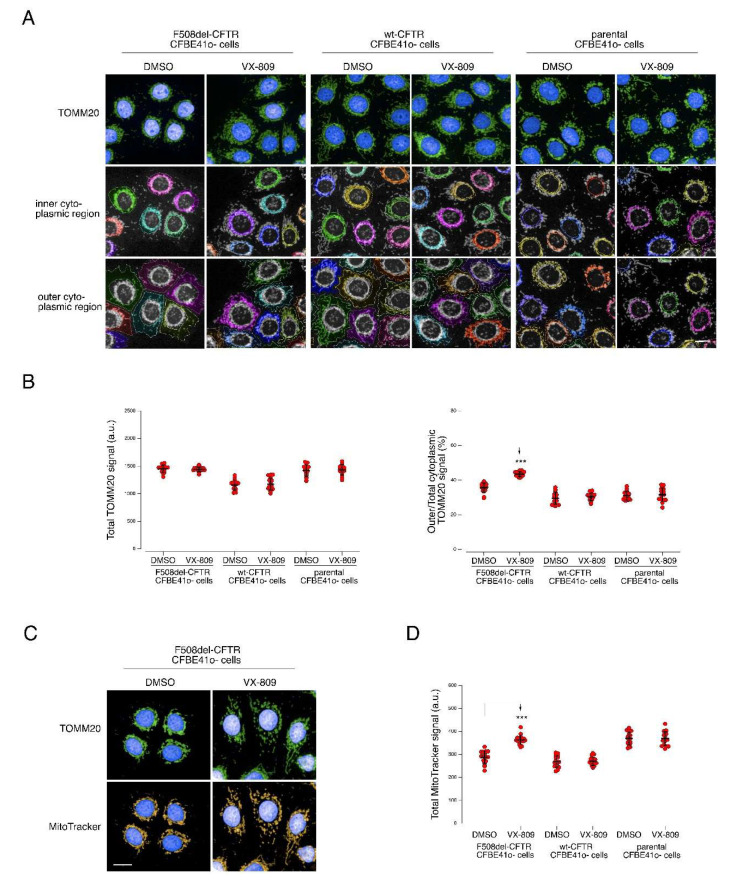
Analysis of mitochondrial network morphology in parental, F508del-CFTR or wt-CFTR CFBE41o- cells under control conditions and following treatment with VX-809. (**A**) Microphotographs showing mitochondrial network in cells treated either with vehicle alone (DMSO) or with VX-809 (3 µM). Mitochondria were visualized by staining for the mitochondrial marker TOMM20 using a mouse anti-TOMM20 as primary antibody followed by an anti-mouse secondary antibody conjugated to AlexaFluor 488. Cell nuclei were counterstained with Hoechst 33342. Cytoplasm was divided into inner (middle panels) and outer (bottom panels) cytoplasmic regions. (**B**) Quantification of total TOMM20 signal intensity (left graph) and percentage of TOMM20 signal localized in the outer cytoplasmic region (right graph) in cells under resting condition (DMSO-treated cells) or treated with VX-809 (3 µM). Each dot represents the value obtained from the analysis of a different biological replicate, in which 1000 cells were imaged and analyzed. Scale bar = 25 µm. Lines indicate means ± SD, *n* = 16. Statistical significance of VX-809 treatment was tested using a Student’s *t*-test. Symbols indicate statistical significance versus DMSO: *** *p* < 0.001. Only statistically significant comparisons are reported. (**C**) Representative microphotographs showing mitochondria in F508del-CFTR CFBE41o- cells treated with vehicle alone (DMSO) or VX-809 (3 µM). Mitochondria were visualized by staining for the mitochondrial marker TOMM20 as detailed in (**A**) or by means of the fluorescent dye MitoTracker, which accumulates in viable mitochondria-dependent on mitochondria membrane potential. Cell nuclei were counterstained with Hoechst 33342. (**D**) Quantification of total MitoTracker signal (corresponding to the dye accumulated into mitochondria) in parental, F508del-CFTR or wt-CFTR CFBE41o- cells treated with DMSO vs. VX-809. Each dot represents the value obtained from the analysis of a different biological replicate, in which 750 cells were imaged and analyzed. Scale bar = 25 µm. Lines indicate means ± SD, *n* = 16. Statistical significance of VX-809 treatment was tested using a Student’s *t*-test. Symbols indicate statistical significance versus DMSO: *** *p* < 0.001. Only statistically significant comparisons are reported.

**Figure 6 cells-11-01938-f006:**
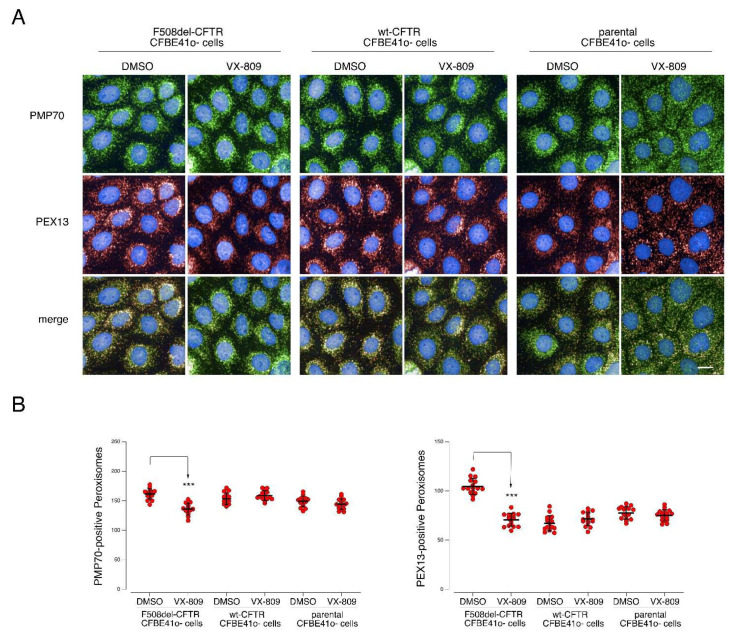
Analysis of peroxisomes distribution in CFBE41o- cells, parental and expressing both F508del and WT-CFTR, under control condition and following treatment with VX-809. (**A**) Microphotographs showing peroxisomes distribution as evidenced by staining for the peroxisomal markers PMP70 and PEX13 in cells treated either with vehicle alone (DMSO) or with VX-809 (3 µM). PMP70 was stained using a rabbit anti-PMP70 as primary antibody followed by an anti-rabbit secondary antibody conjugated to AlexaFluor 488. PEX13 was stained using a mouse anti-PEX13 as primary antibody followed by an anti-mouse secondary antibody conjugated to AlexaFluor 647. Cell nuclei were counterstained with Hoechst 33342. (**B**) Quantification of the number of PMP70- (left graph) and PEX13- positive (right graph) peroxisomes per cell, in cells treated with DMSO vs. VX-809. Each dot represents the value obtained from the analysis of a different biological replicate, in which 700cells were imaged and analyzed. Lines indicate means ± SD, *n* = 16. Statistical significance of VX-809 treatment was tested using a Student’s *t*-test. Symbols indicate statistical significance versus DMSO: *** *p* < 0.001. Scale bar = 25 µm. Only statistically significant comparisons are reported.

**Figure 7 cells-11-01938-f007:**
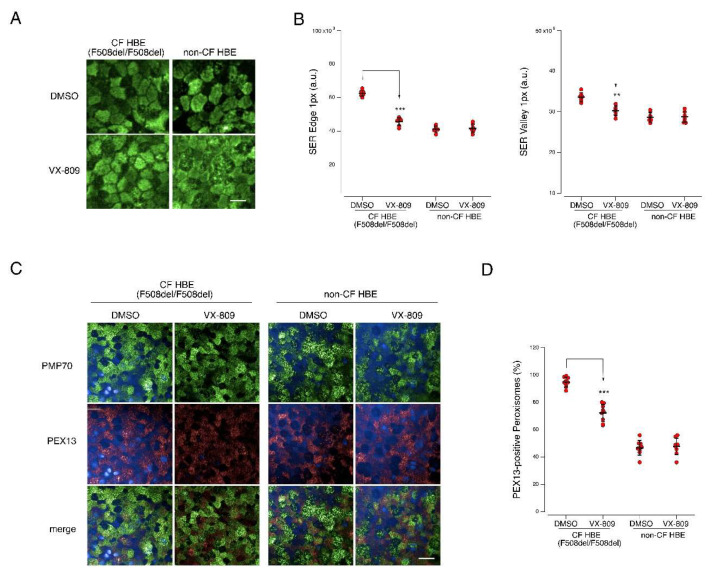
Analysis of mitochondrial network morphology and peroxisomal distribution in CF and non-CF bronchial epithelia. Well-differentiated bronchial epithelia were generated from two F508del/F508del homozygous CF patients (donor ID: BE49 and BE55) and two non-CF subjects (donor ID: BE165 and BE168). (**A**) Microphotographs showing mitochondrial network as evidenced by staining for the mitochondrial marker TOMM20 in epithelia treated for 24 h either with vehicle alone (DMSO) or with VX-809 (3 µM). (**B**) Scatter dot plots showing results of TOMM20 signal texture analysis (based on SER features). Each dot represents the value obtained from the analysis of one region of 1050 µm × 700 µm. Lines indicate means ± SD, *n* = 6. (**C**) Microphotographs showing peroxisomes distribution as evidenced by staining for the peroxisomal markers PMP70 and PEX13 in epithelia treated for 24 h either with vehicle alone (DMSO) or with VX-809 (3 µM). (**D**) Quantification of the calculated percentage of PEX13-positive peroxisomes in each image field. Each dot represents the value obtained from the analysis of one region of 1050 µm × 350 µm. Lines indicate means ± SD, *n* = 9. Statistical significance of VX-809 treatment was tested using a Student’s *t*-test. Symbols indicate statistical significance versus DMSO: *** *p* < 0.001; ** *p* < 0.01. Scale bar = 40 µm. Only statistically significant comparisons are reported.

**Table 1 cells-11-01938-t001:** Proteins with well-defined mitochondrial localization after VX-809 treatment, but not classified as mitochondrial in the control.

Accession	Protein Name	Gene
Q8WVM0	Dimethyladenosine transferase 1, mitochondrial	TFB1M
Q9Y3B7	39S ribosomal protein L11	MRPL11
U3KQ69	Mitochondrial GTPase 1	MTG1
Q969S9	Ribosome-releasing factor 2	GFM2
Q96E29	Transcription termination factor 3	MTERF3
O75616	GTPase Era	ERAL1
Q4G0N4	NAD kinase 2	NADK2
Q5T440	Putative transferase CAF17	IBA57
Q96D53	Atypical kinase COQ8B	COQ8B
Q96DV4	39S ribosomal protein L38	MRPL38
Q8TAE8	Growth arrest and DNA damage-inducible protein	GADD45GIP1
Q96HY7	2-oxoglutarate dehydrogenase E1	DHTKD1
P82933	28S ribosomal protein S9, mitochondrial	MRPS9

**Table 2 cells-11-01938-t002:** List of antibodies for evaluating organelle-specific marker proteins used to evaluate organelle separation via differential centrifugation in LOPIT-DC workflow. All the antibodies listed (with the exception of the antibody to detect the PM marker) were kindly provided by Professor Lilley’s lab at the University of Cambridge (Department of Biochemistry).

Organelle	Antigen	Supplier (Product Number)	Gene Name	MW (kDa)	Source	Dilution for WB
Cytosol	Alpha-Enolase	Cell signalling (3010)	ENO1	47	Rabbit	1:1000
Golgi	Syntaxin-6	Abcam (ab140607)	STX6	32	Rabbit	1:1000
Plasma membrane	Alph-a1 Na^+^/K^+^ ATPase	Abcam (ab76020)	ATP1A1	100	Rabbit	1:5000
ER	Calreticulin	Abcam (ab92341)	CARL	55	Rabbit	1:1000
Chromatin	Histone H2A	Abcam (ab18975)	HIST1H2AG	14	Rabbit	1:1000
Endosome	Early endosome antigen 1	Abcam (ab109110)	EEA1	170	Rabbit	1:1000
Lysosome	Lysosome-associated membrane glycoprotein 1	Cell signalling (3243)	LAMP1	100	Rabbit	1:1000
Nuclear–non-chromatin	Fibrillarin	Cell signalling (2639)	FBL	37	Rabbit	1:1000
Nuclear envelope	Prelamin-A/C	Abcam (ab108922)	LMNA	70	Rabbit	1:1000
Mitochondria	Cytochrome c oxidase subunit 4 isoform 1 (COX 1V)	Cell signalling (4850)	COX4I1	17	Rabbit	1:1000

## Data Availability

The full LOPIT-DC dataset has been made freely explorable at: https://proteome.shinyapps.io/cflopit2021/ (accessed on 30 July 2021). All raw data related to this project (SWATH and LOPIT) are freely available at PRIDE database [61] with accession numbers PXD028355 and PXD028393 respectively.

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
