# Peer review of "CFTR Rescue by Lumacaftor (VX-809) Induces an Extensive Reorganization of Mitochondria in the Cystic Fibrosis Bronchial Epithelium"

_cells, 2022, doi:10.3390/cells11121938_

Round 1

Reviewer 1 Report

In the manuscript “CFTR rescue by lumacaftor (VX-809) induces an extensive reorganization of mitochondria in the cystic fibrosis bronchial epithelium” Clarissa Braccia et al. study the structural cell changes after VX-809 corrector treatment in a HBE41o- cell line with the aim to better understand CFTR drug-mechanism of rescue.

While the results presented in the present article are very interesting, I think that more experimental data should be provided. The manuscript presents a very detailed material and methods section, but there are several shortcomings regarding written style, detailed information of data and graphs. In addition, some errors should be addressed.

-       Several articles are referred only with the doi number. Please, check them and replace with the correct reference format.

-        There are several words and sentences written in cursive, bold type or capital letters (as seen for example in line 153-154 heading) that should be verified.

-        Line 106: add information about which epithelia is described

-        Line 112: add which model are the authors referring (HBEO- cells?)

-        Line 115: “DMSO condition” should include percentage of DMSO administered

-        Line 113: although in the Figure S1 is seen there is a n=5 for each group (“DMSO”, “VX-809”), there is not clear in the results description. Please, clarify if n=5 is for each group. Other points that need clarification:  1) how the control group “QC” is obtained and 2) which “n” is presented in this group (n=10 is mentioned in Line 534, Material & Method section).

-        Along the article, “treated” and “control” is mentioned several times. It can be confused as both DMSO and VX-809 are treatments. I suggest to change it to “VX-809-treated” or “DMSO control” to a clarify.

-      I suggest to do not show repetitive pictures and graphs along the different Figures. Thus, will be graph and figures on top right Figure 1, and pictures and graphs in bottom right of Figure 1. These images/graphs are be confusing here because there is not an extended information and, in my opinion, this information can be more explained in the next figures.

-        Figure S1 – panel A. please increase the font size of DMSO, QC and VX-809 for better visualization. I also suggest to fill the circles of the legend (top right) with colors – the same as they appear in the graph.

-       Figure S1 – panel B. This figure is also unclear. The figure legend does not provide a detailed description of what is showing in the panel. “Treated cells” means VX-809 or DMSO treated? If the panel is showing different conditions, they cannot be differentiated.

-        Increase the font size also in Figure 1.

-    For all immunofluorescence images, colors and antibodies used should be detailed (example: nuclei are shown in blue after Hoechst 33342 staining).

-        Figure S4. Please add more details in figure description.

-      Figure 3B. PM localization of CFTR protein after VX-809 is not well visualized. Please use arrows in the images to indicate the PM localization. Also, it seems that CFTR is more expressed in CFBE410- F508del-CFTR treated with DMSO than CFBE41o- wt treated with DMSO. Do the authors have an explanation?

-        Figure 3D: significant differences are missing in the right graph

-        Figure 3B: the scale bar in the images is missed in the images

-        Figure S5: Increase the font size.

-        Figure legend 3C. Please indicate that this graph shows YFP experiments results.

-      Figure 4. I suggest to change the grey color of numbers for a darker color and with bigger font for better visualization.

-        Figure 4. Please explain the meaning of the numbers between the parenthesis.

-     Figure 5B. The figure description should explain the significances shown in the right graph (as authors did for Figure 5D graph).

-        Line 203. Is it “(76)” a reference or a value?

-        Line 256. Why the forskolin is used at 20 µM? Can you reference a previous article?

-        The use of VX-770 should be clarified:

o   Why VX-770 is added to VX-809 treatment for the short-current circuit when its not added in other experiments?? Please explain why and clarify in which experiments VX-770 is added in the Material & Methods section. Also, it is shown in M & M section than VX-770 is administered in (a) YFP experiment(s?) but this is not explained in the results section (line 231). If the graph right bottom shows short-circuit experiment, I assume VX-770 was used in the “F508del+VX-809” condition. Then, it should be clarified in the graph.

-       Figure 3D: why a CPT-cAMP analogue is used here instead of forskolin, used it in the YPF experiment? Or is the “CPT-cAMP” referred to Forskolin? Please clarify in the using chamber description in the material & methods section.

-        Information from graph in Figure 3D is unclear. It says the graph shown the current after CFTR inhibitor-172? [Itot] is usually refer as total concentration of inhibitor. Then, why wt-CFTR shows CFTR activity? Please clarify this graph.

-    Also clarify when the VX-809 treatment is administered at 3 µM or 1 µM and why different concentrations were used. Please, add references if necessary.

-      About mitochondria staining: As explained in lines 343-344, mitochondria is visualized by both TOMM20 and Mitotracker: please justify the use of 2 different staining and the information that is provided by each one (extend information given in lines 326 – 330 and 352 - 357).

-        Could the authors mention how was performed the quantification after the mitochondria immunofluorescence? (Graphs in Figure 5B and 5D).

-     Lines 373 – 374: it should say “significant reduced” instead of “slightly reduced” as shown in the graph from Figure 6B (left).

-        Figure 6C, D and E is not found in the set of Figures but appear in the description.

-      Lines 400-401. First, authors said one CF patient. But below in line 401 it says two CF patients. Please, clarify.

-       Line 407. Authors declare “significantly different in CF (…) vs non-CF”: 1) does it mean in DMSO condition? And 2) show this significance in the graphs (Figure 7B). In the graphs only significance was shown between DMSO and VX-809 treatment comparing inside the same group.

-        Figure 7C. Blue staining background could be reduced to improve visualization.  

-    Please clarify if CFBE41o- transfected cells were transfected by the authors or were obtained from another lab. References 41 and 42 (line 501) are for YFP information but not for CFBE41o-.  A reference of this cell line should be included.

Minor points:

-        Line 123: should be p>0.05 (as saying no statistical differences) or not necessary.

-        Line 166: please revise “58 markers”

-        Line 176: Should read “PC2 (instead of PC3) vs PC4”, according to Figure S3.

Author Response

Dear Reviewers

We wish to thank you for your inputs and comments, that are helping us in improving the quality of our paper. Please find below in red the point-to-point responses to the points you raised. The corresponding changes to the manuscript and to the supporting data are also indicated in red in the text.

With Best Regards

Andrea Armirotti, Ph.D.

REVIEWER 1

Comments and Suggestions for Authors

In the manuscript “CFTR rescue by lumacaftor (VX-809) induces an extensive reorganization of mitochondria in the cystic fibrosis bronchial epithelium” Clarissa Braccia et al. study the structural cell changes after VX-809 corrector treatment in a HBE41o- cell line with the aim to better understand CFTR drug-mechanism of rescue. While the results presented in the present article are very interesting, I think that more experimental data should be provided. The manuscript presents a very detailed material and methods section, but there are several shortcomings regarding written style, detailed information of data and graphs. In addition, some errors should be addressed.

-      Several articles are referred only with the doi number. Please, check them and replace with the correct reference format. Done

-        There are several words and sentences written in cursive, bold type or capital letters (as seen for example in line 153-154 heading) that should be verified.

-        Line 106: add information about which epithelia is described Done

-        Line 112: add which model are the authors referring (HBEO- cells?) Done

-        Line 115: “DMSO condition” should include percentage of DMSO administered Done

-        Line 113: although in the Figure S1 is seen there is a n=5 for each group (“DMSO”, “VX-809”), there is not clear in the results description. Please, clarify if n=5 is for each group. Other points that need clarification:  1) how the control group “QC” is obtained and 2) which “n” is presented in this group (n=10 is mentioned in Line 534, Material & Method section). We have now clarified this point.

-        Along the article, “treated” and “control” is mentioned several times. It can be confused as both DMSO and VX-809 are treatments. I suggest to change it to “VX-809-treated” or “DMSO control” to a clarify. Done

-      I suggest to do not show repetitive pictures and graphs along the different Figures. Thus, will be graph and figures on top right Figure 1, and pictures and graphs in bottom right of Figure 1. These images/graphs are be confusing here because there is not an extended information and, in my opinion, this information can be more explained in the next figures. Thank you for this suggestion. We reprepared Figure 1. Images and graphs that are repeated in other figures have been removed to remove clutter and improve understanding.

-        Figure S1 – panel A. please increase the font size of DMSO, QC and VX-809 for better visualization. I also suggest to fill the circles of the legend (top right) with colors – the same as they appear in the graph.

I wish we could do this. Unfortunately, the web-based software we used (MEtaboAnalyst) has very limited options for figure preparation (fonts & colors). We have added new axes for the figures on the powerpoint file.

-       Figure S1 – panel B. This figure is also unclear. The figure legend does not provide a detailed description of what is showing in the panel. “Treated cells” means VX-809 or DMSO treated? If the panel is showing different conditions, they cannot be differentiated.

Thanks. This point has been clarified.

-        Increase the font size also in Figure 1. Done.

-        For all immunofluorescence images, colors and antibodies used should be detailed (example: nuclei are shown in blue after Hoechst 33342 staining). Done

-        Figure S4. Please add more details in figure description. Done

-         Figure 3B. PM localization of CFTR protein after VX-809 is not well visualized. Please use arrows in the images to indicate the PM localization. Also, it seems that CFTR is more expressed in CFBE410- F508del-CFTR treated with DMSO than CFBE41o- wt treated with DMSO. Do the authors have an explanation?

The arrows have now been added, as suggested by the reviewer.

CFBE410- expressing F508del-CFTR and CFBE41o- expressing wt-CFTR are different clones both derived from stable transfection with mutant and wild type CFTR protein, respectively, of the parental CFBE41o- cell line. These two cell populations may thus differ in terms of expression of the (heterologous) CFTR protein. Therefore, quantitative comparisons are always performed between different conditions within the same cell line.

-        Figure 3D: significant differences are missing in the right graph Done

-        Figure 3B: the scale bar in the images is missed in the images Done

-        Figure S5: Increase the font size.

-        Figure legend 3C. Please indicate that this graph shows YFP experiments results. Done

-      Figure 4. I suggest to change the grey color of numbers for a darker color and with bigger font for better visualization.

-        Figure 4. Please explain the meaning of the numbers between the parenthesis. I’m not sure I understand this point. Thanks we have now better described these numbers.

-     Figure 5B. The figure description should explain the significances shown in the right graph (as authors did for Figure 5D graph). Done

-        Line 203. Is it “(76)” a reference or a value? It refers to the number of proteins lacking a well-defined relocalization direction. The point has been clarified in the text, now.

-        Line 256. Why the forskolin is used at 20 µM? Can you reference a previous article?

We have added two references for the use of forskolin at 20 µM. Such a concentration was chosen to maximally activate adenylate cyclase.

-        The use of VX-770 should be clarified:

o   Why VX-770 is added to VX-809 treatment for the short-current circuit when its not added in other experiments?? Please explain why and clarify in which experiments VX-770 is added in the Material & Methods section. Also, it is shown in M & M section than VX-770 is administered in (a) YFP experiment(s?) but this is not explained in the results section (line 231). If the graph right bottom shows short-circuit experiment, I assume VX-770 was used in the “F508del+VX-809” condition. Then, it should be clarified in the graph.

VX-770 is a potentiator that increases open channel probability, that was used only in functional assay to maximally activate CFTR channel. Thus, it has been used on all functional assay performed on F508del-CFTR CFBE41o- cells (both DMSO- or VX-809-treated) and on wt-CFTR CFBE41o- cells (where it exerted negligible effects). The legend for figure 3D clearly reports that “the dot plot shows the total current sensitive to inh-172 (ITOT) measured in each epithelium, used as parameter of CFTR function”. It also reports that “cells were stimulated with the membrane permeable cAMP analogue CPT-cAMP (100 µM) followed by VX-770 (1 µM) to maximally activate F508del-CFTR. Then, CFTR currents were blocked adding the CFTR inhibitor-172 (inh-72, 10 µM).” As it can be seen in the representative short-circuit current traces, VX-770 was applied acutely during the recordings performed on epithelia derived from both CF and non-CF donors.

-       Figure 3D: why a CPT-cAMP analogue is used here instead of forskolin, used it in the YPF experiment? Or is the “CPT-cAMP” referred to Forskolin? Please clarify in the using chamber description in the material & methods section.

It is common practice to use a membrane-permeable, not-hydrolysable cAMP analogue (instead of the adenylate cyclase activator forskolin) to experimentally increase cAMP content in primary airway epithelia. This is done mainly to bypass the enzymatic step that leads to cAMP content increase and thus to decrease experimental variability.

-        Information from graph in Figure 3D is unclear. It says the graph shown the current after CFTR inhibitor-172? [Itot] is usually refer as total concentration of inhibitor. Then, why wt-CFTR shows CFTR activity? Please clarify this graph.

ITOT stands for total current and not for total concentration of inhibitor. Indeed, during the experiments epithelia were stimulated with the membrane permeable cAMP analogue CPT-cAMP followed by VX-770 to

maximally activate CFTR. Then, CFTR currents were blocked adding the CFTR inhibitor-172. The amplitude of the current drop caused by inh-172 was taken as an estimate of total CFTR activity (ITOT) in each epithelium. Epithelia expressing wt-CFTR show a greater CFTR-mediated current than epithelia expressing mutant CFTR.

-    Also clarify when the VX-809 treatment is administered at 3 µM or 1 µM and why different concentrations were used. Please, add references if necessary.

Considering the dose-response relationship for VX-809 as mutant CFTR corrector, 1 µM is already a concentration that achieves maximal rescue (DOI: 10.1126/sciadv.aay9669). A sentence has been added to the M & M section.

-      About mitochondria staining: As explained in lines 343-344, mitochondria is visualized by both TOMM20 and Mitotracker: please justify the use of 2 different staining and the information that is provided by each one (extend information given in lines 326 – 330 and 352 - 357).

We used two different stainings to address mitochondrial morphology and function/viability. Indeed, TOMM20 staining was used to assess mitochondrial network morphology. Mitochondrial activity was instead measured by means of MitoTracker Red, a fluorescent dye whose accumulation is dependent upon mitochondria membrane potential.

-        Could the authors mention how was performed the quantification after the mitochondria immunofluorescence? (Graphs in Figure 5B and 5D).

As described in the M & M section, image analysis of signal intensity and morphology, as well as automatic detectionof signal spots (resembling peroxisomes), were performed using the Harmony software (version 4.9) of the Opera Phenix high-content system. This software automatically detects and quantifies the presence of a fluorescent signal, and its intensity and morphology, based on  machine-learning algorithms (PhenoLogic).

-     Lines 373 – 374: it should say “significant reduced” instead of “slightly reduced” as shown in the graph from Figure 6B (left).

We removed the word “slightly”

-        Figure 6C, D and E is not found in the set of Figures but appear in the description. Thanks, the legend has been fixed.

-      Lines 400-401. First, authors said one CF patient. But below in line 401 it says two CF patients. Please, clarify. Primary cells from two CF and two non-CF donors were used, as described. Text has been fixed at line 401.

-       Line 407. Authors declare “significantly different in CF (…) vs non-CF”: 1) does it mean in DMSO condition? And 2) show this significance in the graphs (Figure 7B). In the graphs only significance was shown between DMSO and VX-809 treatment comparing inside the same group. Thanks, this point has been clarified.

-        Figure 7C. Blue staining background could be reduced to improve visualization.  As requested by the reviewer, we have reduced blue staining background to improve visualization.

-    Please clarify if CFBE41o- transfected cells were transfected by the authors or were obtained from another lab. References 41 and 42 (line 501) are for YFP information but not for CFBE41o-.  A reference of this cell line should be included.

Reference 42 clearly describe how the CFBE41o- cell lines were obtained: “The CF bronchial epithelial cells CFBE41o-, with stable expression of F508del-CFTR or wild type CFTR… obtained by Dr. J.P. Clancy…. were transfected with the halide-sensitive yellow fluorescent protein (HS-YFP) YFP-H148Q/I152L”

Minor points:

-        Line 123: should be p>0.05 (as saying no statistical differences) or not necessary. Done

-        Line 166: please revise “58 markers”. I don’t understand the point. Of the 548 manually curated markers, 58 refer to cytosol proteins, 96 to ER proteins…etc etc.

-        Line 176: Should read “PC2 (instead of PC3) vs PC4”, according to Figure S3. Fixed, thanks.

Reviewer 2 Report

The authors aimed at investigating the effect of CFTR modulator VX-809 on the proteome of F508del-CFTR expressing CFBE cells. While they did not observe significant changes in protein expression, they found changes in the localization and structure of mitochondria and peroxisomes after 24h treatment with VX-809. The changes seem to be directly linked to the rescue of F508del-CFTR at the PM as those were not observed in wt-CFTR expressing or parental CFBE cells when treated with VX-809. Mitochondrial morphology and peroxisomal distribution after VX-809 treatment was confirmed in primary CF cells. The work is interesting and shows additional effects of CFTR modulator VX-809 on cell organelle structures and protein localization beyond its corrector function on F508del-CFTR folding. However, there are still some aspects that should be clarified.

Why was VX-809 used at different concentrations (1µM or 3µM). The lower concentration could be a reason for no/minor effects in the proteomic studies.

Figure 3:

F508del-CFTR expressing cells after treatment with VX-809 appear to have more CFTR expression compared to wt-CFTR expressing cells in the IF images.

Figure 5A and B:

One can clearly see a difference in the total TOMM20 staining between F508del-CFTR DMSO and VX-809 treated cells while the authors state there is not. The image chosen is then not representative.

Figure 5C, 5D:

Why are the nuclei white in the images of DMSO treatment?

Why is TOMM20 signal unchanged compared to MitoTracker. One cannot see that there is any difference between the staining with TOMM20 and MitoTracker. If it is about signal intensity it should be better described.

Figure 6:

The reduction of peroxisome marker PMP70 is not obvious in the representative image while it is clearly for PEX13.

Figure 7 A and C:

It is not well described what is shown here. Is this a maximum/average intensity projection of image stacks? Fig7A: Upper left and lower right images are very blurry and should be replaced. It is not labelled that cells are stained with mitochondrial marker TOMM20. Fig7C: Image quality should be improved. Some images also appear very blurry.

Minor comments

Line 81:

Reference needs proper insertion into bibliography

Line 460:

It is the rescue of F508del-CFTR triggered by VX-809

Discussion

Here, it should be clearly mentioned that changes after treatment with VX-809 were seen in F508del-CFTR expressing CFBE41o- cells and not only CFBE41o- cells, which rather refers to the parental cell line were the authors did not see changes after treatment.

Author Response

Dear Reviewers

We wish to thank you for your inputs and comments, that are helping us in improving the quality of our paper. Please find below in red the point-to-point responses to the points you raised. The corresponding changes to the manuscript and to the supporting data are also indicated in red in the text.

With Best Regards

Andrea Armirotti, Ph.D.

REVIEWER 2

The authors aimed at investigating the effect of CFTR modulator VX-809 on the proteome of F508del-CFTR expressing CFBE cells. While they did not observe significant changes in protein expression, they found changes in the localization and structure of mitochondria and peroxisomes after 24h treatment with VX-809. The changes seem to be directly linked to the rescue of F508del-CFTR at the PM as those were not observed in wt-CFTR expressing or parental CFBE cells when treated with VX-809. Mitochondrial morphology and peroxisomal distribution after VX-809 treatment was confirmed in primary CF cells. The work is interesting and shows additional effects of CFTR modulator VX-809 on cell organelle structures and protein localization beyond its corrector function on F508del-CFTR folding. However, there are still some aspects that should be clarified.

 Why was VX-809 used at different concentrations (1µM or 3µM). The lower concentration could be a reason for  no/minor effects in the proteomic studies.

Considering the dose-response relationship for VX-809 as mutant CFTR corrector, 1 µM is already a concentration that achieves maximal rescue (DOI: 10.1126/sciadv.aay9669). A sentence has been added to the M & M section.

Figure 3:

F508del-CFTR expressing cells after treatment with VX-809 appear to have more CFTR expression compared to wt

CFTR expressing cells in the IF images.

CFBE410- expressing F508del-CFTR and CFBE41o- expressing wt-CFTR are different clones both derived from stable transfection with mutant and wildtype CFTR protein, respectively, of the parental CFBE41o- cell line. These two cell populations may thus differ in terms of expression of the (heterologous) CFTR protein. Therefore, quantitative comparisons are always performed between different conditions within the same cell line.

Figure 5A and B:

One can clearly see a difference in the total TOMM20 staining between F508del-CFTR DMSO and VX-809 treated

cells while the authors state there is not. The image chosen is then not representative.

There is not a difference in the total TOMM20 signal between F508del-CFTR DMSO and VX-809 treated cells. There is a difference in the “distribution” of the TOMM20 signal between F508del-CFTR DMSO and VX-809 treated

cells (that can be easily seen by eye – Fig. 5B right graph), but not in the Total signal, which comprises both signal distribution (area) and intensity (Fig. 5B left graph).

Figure 5C, 5D:

Why are the nuclei white in the images of DMSO treatment?

We have maintained the same settings while acquiring all the confocal images. In these images the blue signal is saturated, thus it appears white. This however does not interfere with the automatic quantification of signal intensity and distribution for the other fluorophores that have different wavelengths. We have decided to substitute the images to avoid showing saturated nuclear signals. Please see revised Figure 5.

Why is TOMM20 signal unchanged compared to MitoTracker. One cannot see that there is any difference between

the staining with TOMM20 and MitoTracker. If it is about signal intensity it should be better described.

These two different stainings address mitochondrial morphology and function/viability. Indeed, TOMM20 staining was used to assess mitochondrial network morphology. Mitochondrial activity was instead measured by means of MitoTracker Red, a fluorescent dye whose accumulation (and thus, signal intensity) is dependent upon mitochondria membrane potential.

Figure 6:

The reduction of peroxisome marker PMP70 is not obvious in the representative image while it is clearly for PEX13.

Indeed, it is difficult to detect by eye a reduction of 12% in a fluorescent signal, as in the case of PMP70. On the contrary, the reduction in PEX13 accounts for more than 40%, thus it can easily be seen by eye.

Figure 7 A and C:

It is not well described what is shown here. Is this a maximum/average intensity projection of image stacks? Fig7A: Upper left and lower right images are very blurry and should be replaced. It is not labelled that cells are stained with mitochondrial marker TOMM20. Fig7C: Image quality should be improved. Some images also appear very blurry.

We agree with the reviewer about the poor quality of HBE images, and we have done our best to choose the most representative and suitable images for this revised version of the manuscript. However, although the Opera Phenix (in particular, our instrument) is based on a confocal optics, it has a water-immersion 40X objective, thus it is designed not to acquire “beautiful images” but to acquire “highly informative” images. Imaging of intact epithelia with the Opera Phenix is particularly difficult and the images look like being of poor quality, and also the detection of cell borders is not easy. On the other side, this automated system allows to screen and thus to generate data based on signal pattern recognition also from images like these. Images in figure 7A and C are single plane images, not maximum intensity projections of image stacks. In addition to this, it has to be considered that the epithelium is not perfectly planar, thus by focusing in the center of the image, usually the borders appear “blurry”. Despite the apparent “poor quality” of the images, the Opera Phenix allows to analyze in details signal morphology, intensity and texture, by means of the PhenoLogic machine-learning algorithms of the Harmony software of the system.

As for Figure 7A, in the legend it is reported that the images show the staining for the mitochondrial marker TOMM20. Usually, the images are labelled when there are different markers shown and compared in the images.

Minor comments

Line 81:

Reference needs proper insertion into bibliography

Done

Line 460:

It is the rescue of F508del-CFTR triggered by VX-809. It is not clear what the reviewer is referring to.. VX-809 is a compound able to bind to mutant CFTR protein and improve its folding, to promote its trafficking to the plasma membrane.

Discussion

Here, it should be clearly mentioned that changes after treatment with VX-809 were seen in F508del-CFTR expressing

CFBE41o- cells and not only CFBE41o- cells, which rather refers to the parental cell line were the authors did not see

changes after treatment.

As suggested by the reviewer, we have clearly specified in the Discussion that changes after treatment with VX-809 were seen in F508del-CFTR expressing CFBE41o- cells.